# Aerosol trace element solubility and deposition fluxes over the Mediterranean and Black Sea basins

Rachel U. Shelley [1], Alex R. Baker [1], Max Thomas [1,2], Sam Murphy [1,3]

1 – Centre for Ocean and Atmospheric Sciences, School of Environmental Sciences, University of East Anglia, Norwich, NR4 7TJ, UK

2 – now at: Met Office Hadley Centre, Exeter, EX1 3PB, UK

3 – now at: Hydrock now Stantec, Merchants' House North, Wapping Road, Bristol, BS1 4RW, UK

*Correspondence to*: Alex Baker (alex.baker@uea.ac.uk)

**Abstract.** Aerosol samples collected during summer 2013 on GEOTRACES cruise GA04 in the Mediterranean and Black seas were analysed for their soluble and total metal and major ion composition. The fractional solubilities (soluble / total concentrations) of the lithogenic elements (Al, Ti, Mn, Fe, Co, Th) varied strongly with atmospheric dust loading. Solubilities of these elements in samples that contained high concentrations of mineral dust were noticeably lower than at equivalent dust concentrations over the Atlantic Ocean. This behaviour probably reflects the distinct transport and pollutant regimes of the Mediterranean basin. Elements with more intense anthropogenic sources (P, V, Ni, Cu, Zn, Cd, Pb) had a variety of largely independent sources in the region and generally displayed higher fractional solubilities than the lithogenic elements. Calculated dry deposition fluxes showed a west to east decline in the N/P ratio in deposition over the Mediterranean, a factor that contributes to the P-limited status of the eastern basin. Atmospheric deposition may make a significant contribution to the surface water budgets of Mn and Zn in the western Mediterranean.

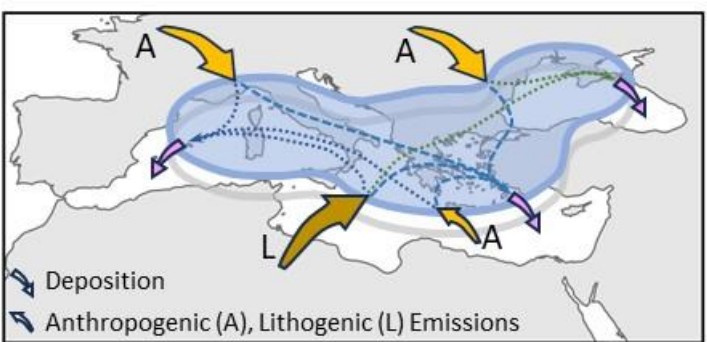

## 1 Introduction

Aerosols have both direct and indirect impacts on the global climate system (e.g. Martin, 1990; Carslaw et al., 2010). Aerosols directly interact with incoming and outgoing solar radiation and act as seeds for cloud formation. Indirectly, they impact the climate through the deposition of both essential and potentially toxic elements to aquatic and terrestrial ecosystems, where

they can be assimilated by photosynthetic microbes (Jickells et al., 2005; Jickells and Moore, 2015; Okin et al., 2011; Westberry et al., 2023).

Aerosol metals are sourced from natural (mineral dust, sea spray, biomass burning/wildfires, glacial flour, volcanoes, bioaerosols) and anthropogenic (metal smelting, industry, mining, vehicles and shipping) sources, and can be transported vast distances in the atmosphere. In marine environments, sea spray aerosol is often the largest contributor to aerosol mass, especially far from anthropogenic and continental sources (de Leeuw et al., 2011), although mineral dust arguably exerts a greater impact on ecosystem functioning, specifically with respect to the input of biogeochemically important elements (Jickells et al., 2005; Hamilton et al., 2022). In marine regions under the transport pathways of mineral dust from the world's major desert regions, atmospheric deposition is a major external source of macro- (P) and micronutrients (e.g., Fe, Mn, Cu, Zn) (Hamilton et al., 2022).

The Mediterranean and Black seas are isolated from the global ocean system: the Mediterranean being connected to the Atlantic Ocean via the Strait of Gibraltar, with the Mediterranean and Black seas connected via the Dardanelles-Sea of Marmara-Bosporus system. During the summer, both seas are highly stratified, largely due to thermal stratification in the Mediterranean and also due to high riverine inputs of fresh water in the Black Sea. Under these conditions, the main external sources of nutrients to the surface waters of both seas are atmospheric and riverine inputs. The Mediterranean is classified as a Low Nutrient Low Chlorophyll (LNLC) ecosystem, characterised by increasing oligotrophy from the western to eastern basins. The Black Sea is the world's largest permanently anoxic marine basin, a status that profoundly shapes its ecosystem functioning. Compared to open-ocean waters, high concentrations of dissolved trace metals (e.g., Al, Mn, Fe, Co, Ni, Cu, Zn, Cd, Pb) are observed in the surface waters of both the Mediterranean and Black seas (Boyle et al., 1985; Guieu et al., 1998; Tovar-Sanchez et al., 2014; Dulaquais et al., 2017; Gerringa et al., 2017; Middag et al., 2022). Both seas are heavily impacted by the atmospheric deposition of natural and anthropogenic material due to their proximity to the Sahara Desert, Europe, and the mega-cities of Cairo and Istanbul (Kubilay et al., 1995; Koçak et al., 2007; Heimburger et al., 2010; Theodosi et al., 2010b; Theodosi et al., 2013; Kanakidou et al., 2020). Seasonal wildfires (Hernandez et al., 2015) and intense shipping activity in the Mediterranean (Becagli et al., 2012) also add to the aerosol load in the region. In the summer, boundary layer air over the Mediterranean is heavily polluted by anthropogenic sources in Western and Eastern Europe, with occasional incursions of mineral dust from North Africa (Lelieveld et al., 2002; Gkikas et al., 2013). African-origin dust also occasionally reaches the Black Sea in summer (Kubilay et al., 1995).

The trace metal composition of aerosols over the Mediterranean has been reported to be dominated by the mixing of these anthropogenic and natural sources (Chester et al., 1981; Chester et al., 1993; Herut et al., 2001). The impact of atmospheric deposition of aerosol-associated elements on the biogeochemistry of the Mediterranean-Black Sea system remains poorly understood (Guieu et al., 2010a; Guieu et al., 2010b), in part because the factors that influence trace element fractional solubility (and hence availability to the microbial community) in this system are unclear. For several dust-borne trace elements (some of which are biogeochemically important, e.g. Fe, Cu) baseline solubility appears to be low but increases during atmospheric transport (Jickells et al., 2016; Shelley et al., 2018), as a result of interactions of mineral dust with acidic gases

($SO_2$ and $NOx$), mixing of dust with trace elements of anthropogenic origin and other processes (Baker and Croot, 2010). These processes are likely to operate rather differently in the Mediterranean / Black Sea atmosphere compared to the relatively well studied Atlantic atmosphere (e.g. Buck et al., 2010; Jickells et al., 2016; Shelley et al., 2018), due to the differences in the relative proportions of dust and anthropogenic emissions and their transport and mixing regimes.

In this paper, we discuss data for a suite of total and soluble elements from aerosols collected during the GEOTRACES cruise, GA04, to the Mediterranean and Black seas. The solubility of trace elements primarily associated with mineral dust (Al, Ti, Mn, Fe, Co, Th) is considered separately from elements whose sources are more closely associated with anthropogenic activities (P, V, Ni, Cu, Zn, Cd, Pb), although individual aerosol samples will contain a mixture of both lithogenic and anthropogenic sources of each element (e.g. Chester et al., 1993). The potential influences on the solubility of these elements in the unusual conditions of the Mediterranean / Black Sea atmosphere, where the surrounding land masses contain significant sources of mineral dust and anthropogenic pollutants is discussed. The potential impact of atmospheric deposition on surface water macronutrient (N/P) status and dissolved metal budgets of the Mediterranean Sea is also considered.

## 2 Methods

### 2.1 Study region

Aerosol samples were collected during the GEOTRACES cruise GA04 aboard the RV *Pelagia* during May-August 2013 (Table 1, Fig. 1). A full description of the Mediterranean, Aegean and Sea of Marmara sections of the cruise track (Legs 1 and 3) has previously been published (Rolison et al., 2015; Dulaquais et al., 2017; Gerringa et al., 2017; Middag et al., 2022).

Table 1. Summary of the GA04 cruise legs, dates in 2013 and sample numbers.

| | Cruise Leg | Dates | Samples |
|---|---|---|---|
| Leg 1 | Southern leg, Lisbon – Istanbul | 16 May – 4 June | 01 – 18 |
| Leg 2 | Black Sea, Istanbul – Istanbul | 14 - 22 July | 19 - 25 |
| Leg 3 | Northern leg, Istanbul – Lisbon | 26 July – 10 August | 26 - 40 |

### 2.2 Sample collection

Two high volume, mass flow controlled, Total Suspended Particulate aerosol samplers (Tisch Environmental, Ohio, USA) were located on the wheelhouse roof. Samples were collected on Whatman 41 (W41) filter sheets (203 x 254 mm, cellulose). For trace metals (TM), filters were acid washed before use (0.5 M HCl and 0.1 M $HNO_3$; Baker et al., 2007), while for major ions (MI), filters were used without pre-treatment. The aerosol samplers were automatically controlled to operate during relative (from the ship's bow) wind directions of between -80 ° and 145 ° and at wind speeds $> 2$ m s$^{-1}$ to minimise the risk of

contamination from the ship's exhaust (Chance et al., 2015). TM and MI samples were generally collected in pairs, although the MI sampler malfunctioned during Leg 2. No MI data are available for this section of GA04. Samples were collected over ~24 h periods, with the exception of sample TM19 which ran for 46 h. Three types of filter blanks were collected: 1) cassette blank – filter placed in filter cassette and stored in a plastic bag for 24 h, 2) process blank – filter placed in filter cassette and placed in the aerosol sampler and left for 24 h without the motor running, 3) motor blank – filter placed in filter cassette and placed in the aerosol sampler with the motor running for 5 min. Upon collection, all filter samples were folded inwards, zip-lock bagged and stored frozen (-20 °C) onboard the ship. Samples and blanks remained frozen in the home laboratory until either strong acid digestion (total trace metals) or leaching for soluble trace metals or major ions. Reagent blanks were also taken to monitor for potential contamination arising from the digestion or leaching processes. Where blanks were above the analytical limit of detection for an analyte, these were averaged and subtracted from the results obtained for the samples. If blanks were below the limit of detection no blank subtraction was done.

## 2.3 Sample analysis

### 2.3.1 Total trace elements (Al, P, Ti, V, Mn, Fe, Co, Ni, Cu, Zn, Cd, Pb, Th)

One eighth of the TM filter was placed in a 15 mL Teflon vial. Four mL of a 4:1 mixture of ultrapure $HNO_3$: HF (both Merck, Trace Select Ultra) was pipetted in, and the vial capped. The samples were digested using a microwave system (Milestone UltraWave) using the silica sand pre-programmed method (T = 240 °C, P = 110 bar). Following the digestion, the solutions were transferred to clean Teflon vials and taken to near-dryness on a Teflon-coated hotplate. The hotplate was housed in a fume cupboard, inside a plastic box, from which acid fumes were drawn through a saturated solution of $CaCO_3$ in order to remove excess HF (Baker et al., 2020). The residue was then redissolved in 0.48 M $HNO_3$ (Shelley et al., 2015; Shelley et al., 2017). All samples and blanks were subsequently spiked with 10 ppb Rh as an internal standard. Aluminium, Fe and P were determined by ICP-AES (Thermo Scientific i-CAP PRO). All other elements (Ti, V, Mn, Co, Ni, Cu, Zn, Cd, Pb, Th) were determined by ICP-MS (Thermo Scientific, iCAP TQ) and calibrated using external standards (SPEX CertiPrep). Calibrations were verified by analysis of matrix reference materials (MRMs TM-27.3, TMDA-64.2, TM-27.4 and TMDA-62.3; Environment Canada) with recoveries being within 5% (Al, Ti, V, Mn, Fe, Zn), 10% (Co, Ni, Cd) and 15% (Cu, Pb) of certified values. Aliquots of Arizona Test Dust were also digested alongside aerosol samples. The results obtained are summarised in Table S1. Total Cr, As, Sb, Ba, La, Ce, Nd and U were also determined but are not discussed here as there are no corresponding soluble metal data for these elements.

### 2.3.2 Soluble trace elements (Al, Ti, V, Mn, Fe, Co, Ni, Cu, Zn, Cd, Pb, Th)

One quarter of each TM filter was leached for 1-2 h with ammonium acetate buffered to pH 4.7, with occasional gentle shaking by hand. The resulting leachate was filtered using 0.2 µm pore sized cellulose acetate syringe filters (minisart, Sartorius) (Baker

et al., 2007). The samples were analysed by ICP-AES (Varian Vista Pro, Fe, Al, Mn, Ti, Zn, V) and ICP-MS (Thermo X-Series; Cu, Ni, Co, Cd, Pb, Th). Calibrations were prepared from external standards (SPEX CertiPrep) and recoveries for MRMs (TM-27.3, TMDA-64.2; Environment Canada) were within 5% (Ti, Mn, Fe, Co, Ni, Cu, Pb), 10% (Al, V, Zn) and 25% (Cd) of certified values.

### 2.3.3 Soluble major ions ($Na^+$, $NH_4^+$, $Mg^{2+}$, $K^+$, $Ca^{2+}$, $Cl^-$, $NO_3^-$, $SO_4^{2-}$, $Br^-$, $C_2O_4^{2-}$) and soluble phosphate

One quarter of each MI filter was leached using ultrahigh purity water (18.2 MΩ.cm) with ultrasonic agitation for 1 hour (Baker et al., 2007). After filtration (0.2 µm), samples were analysed using a Dionex ICS-5000 dual channel ion chromatograph equipped with CS12A and AS18 columns for cation and anion separation, respectively. Calibration standards were prepared from analytical grade salts (Fisher Scientific) of the ions and verified against MRMs (ION-915 and KEJIM-02; Environment Canada). Recoveries for these MRMs were within 5% ($Na^+$, $Mg^{2+}$, $Ca^{2+}$, $Cl^-$, $NO_3^-$, $SO_4^{2-}$) and 10% ($K^+$) of their

certified values. For soluble phosphate determination, one eighth of each MI filter was leached using 1 mM $NaHCO_3$, also using 1 hour of ultrasonic agitation. Soluble reactive phosphorus (hereafter referred to as phosphate) was determined in the filtered leachates by spectrophotometry (Baker et al., 2007).

### 2.4 Calculation of derived parameters

Fractional solubility was calculated as the ratio s-X / t-X, expressed as a percentage, where s-X and t-X are the soluble and
total concentrations respectively for each trace element (X).

Enrichment factors (EF) were calculated using the elemental ratios of the element of interest (X) and Al (as the lithogenic tracer) in aerosol with respect to the X/Al ratio in shale (Turekian and Wedepohl, 1961), according to: EF = $([X]_{aero}/[Al]_{aero})/([X]_{shale}/[Al]_{shale})$. Elements of predominantly lithogenic origin are not expected to be enriched and to have EFs close to 1. As there is natural variability in the abundances of elements in crustal material, we only consider EFs > 10 as

significantly enriched, with other (e.g. anthropogenic) sources of these elements dominating over lithogenic sources.

The non-seasalt (nss) component of MI concentrations were calculated by subtracting the contribution of the ion arising directly from seaspray aerosol from the measured concentration (nss-MI = $[MI_{aero}]$ - $[MI_{ss}]$). Seaspray aerosol contributions were estimated from measured aerosol $Na^+$, assuming that the ratio of the ion to $Na^+$ in seaspray is identical to that of bulk seawater (Baker et al., 2007).

### 140 2.5 Dry deposition fluxes

The dry deposition flux (F) was calculated from the product of the concentration of that element ($C_X$) and a deposition velocity ($v_d$); F = $C_x$ * $v_d$. Because it is not possible to accurately determine the deposition velocity over the ocean, there is large (a factor of 2-3) uncertainty associated with the $v_d$ term (Duce et al., 1991). For our flux calculations, we used $v_d$ values of 1 cm $s^{-1}$ for the lithogenic elements and 0.1 cm $s^{-1}$ for the anthropogenic elements, in order to reflect the size distribution of the
particles where these elements are found (Duce et al., 1991). When estimating deposition to the Mediterranean basin we

assumed its surface area to be 2.5 x $10^6$ km$^2$, and the relative areas of the western and eastern basins to be 1:2. Mineral dust deposition fluxes were estimated from t-Al, assuming that Al is 8 weight percent of dust (Turekian and Wedepohl, 1961).

## 2.6 Air mass back trajectories

Five-day air mass back trajectory (AMBT) simulations were produced using the NOAA Air Resources Laboratory's HYPLIT
model (https://www.ready.noaa.gov/HYSPLIT_traj.php) with NCEP/NCAR Reanalysis Project datasets (Stein et al., 2015). Trajectories were calculated for arrival heights of 10, 500 and 1000 m above mean sea level at 3 hourly intervals along the ship's track. Based on the AMBTs, samples were assigned to one of five air mass types, indicative of likely aerosol source characteristics as described below. Assignments were done principally based on the surface level (10 m) trajectories. However, higher altitude transport (up to 3000 m) can be significant for Saharan dust over the Mediterranean (Scerri et al., 2016), so
upper levels were also considered in the context of transport from North Africa.

## 3 Results and Discussion

### 3.1 Air mass origins during GA04

Example AMBTs and air mass type assignments are shown in Fig. 2 (and in more detail in Fig. S1). Many of the air masses encountered during GA04 arrived at the ship from a north-westerly direction, having passed over continental Europe. We
divided these European air masses into two categories. The Eastern European (EEU, n = 15) group had most recently been in contact with Europe to the east and north of the Adriatic Sea, while the Western European (WEU, n = 10) group includes arrivals most recently in contact with mainland Italy, France and the Iberian Peninsula. The Mediterranean (MED, n = 4) and Remote North Atlantic (RNA, n = 3) groups had not interacted with continental land masses within the five days of the simulation (although MED trajectories that passed over islands were not uncommon and for RNA sample TM03 trajectories
passed briefly over southwestern regions of Iberia: see Fig. S1). Eight samples were affected by air mass arrivals from North African (NAF). As illustrated in Fig. 2, trajectories for this group passed over regions identified as potential source areas (PSAs) for dust aerosols, notably the 'Libya/Egypt' and 'Libya/Algeria/Mali' PSAs (Guinoiseau et al., 2022), but we do not attempt to assign potential dust sources for individual samples. All of the NAF samples showed the orange colouration associated with desert dust.


### 3.2 Total lithogenic elements (t-Al, Ti, Mn, Fe, Co, Th)

For all the primarily lithogenic elements, the influence of Saharan dust (NAF air mass type) on the aerosol total metal concentrations is clearly visible in samples TM8-14 (Fig. 3, Table S2). The NAF samples had an orange colour, typical of aerosols with Saharan origin, and all of the lithogenic elements had their maximum concentrations within this air mass type.
Molar ratios to Al for these elements (Table S3) were reasonably consistent across the GA04 Mediterranean-Black Sea air

mass types and with their ratios reported in Saharan aerosols collected over the tropical North Atlantic (Shelley et al., 2015; Jickells et al., 2016) and in the eastern Mediterranean (Herut et al., 2016). As expected for elements from primarily crustal sources, no significant enrichment was observed (i.e. EF ~ 1) for the lithogenic elements, for all air mass types. (Very similar behaviour was also observed for Ba, La, Ce, Nd and U during GA04 (Fig. S2)). In a few cases EFs for Mn and Co were >3 in European air masses (Fig. 3c & e), but Co was the only lithogenic element to be notably enriched (EF >10) in samples (TM03 and 04) whose trajectories passed over the southern tip of Iberia. Lithogenic element concentrations and EF values from GA04 are consistent with previous observations from shipboard sampling (Tables S4 & S5; Chester et al., 1993; Chester et al., 1984; Moreno et al., 2010; Hacisalihoglu et al., 1992; Kubilay et al., 1995) and at a number of coastal and island time series sites (e.g. Kubilay and Saydam, 1995; Koçak et al., 2007; Guieu et al., 2010b; Heimburger et al., 2010).

### 3.3 Soluble lithogenic elements (s-Al, Ti, Mn, Fe, Co, Th)

In contrast to the total metal concentrations, the highest soluble concentrations for the lithogenic elements were not found in samples from NAF air masses. For example, median s-Fe concentrations in aerosols of the Mediterranean and European air mass types (110, 320 and 180 pmol m$^{-3}$ for MED, WEU, EEU types respectively) were all considerably higher than in NAF aerosols (54 pmol m$^{-3}$) during GA04. The median s-Fe concentration in the RNA air mass was ~30 pmol m$^{-3}$, similar to the value previously reported for this air mass type in the remote Atlantic: ~15 pmol m$^{-3}$ (Baker and Jickells, 2017). This was probably a result of the lower fractional solubility of the lithogenic elements in mineral dust compared to aerosols from other sources (Fig. 4 and Jickells et al., 2016; Shelley et al., 2018) combined with higher fractional solubility of these elements in anthropogenic sources (e.g. median fractional solubilities for Fe in the NAF, MED, WEU, EEU and RNA types during GA04 were 0.4, 2.9, 12.4, 2.8 and 5.1 %, respectively) and the relatively high (compared to the atmosphere over the North Atlantic) concentration of anthropogenic aerosols encountered during GA04. A common feature of the soluble lithogenic elements during GA04 was the two peaks in concentration in samples TM18 (Sea of Marmara) and TM20 (SW Black Sea), both from EEU air masses, which corresponded with peaks in fractional solubility (Fig. 4).

### 3.4 Total anthropogenic elements (t-P, V, Ni, Cu, Zn, Cd, Pb)

In contrast to the lithogenic elements, the distributions of the anthropogenic elements during GA04 were more diverse (e.g. they had different concentration profiles along the cruise track and in most cases their highest concentrations were not associated with the NAF samples, Fig. 5). All of the anthropogenic elements displayed significant enrichment (EF >10) relative to crustal material, although enrichments were lowest, and in some cases (e.g. for t-V and t-Ni) close to 1, for the NAF samples. These distributions are consistent with these elements having a variety of disparate sources around the region (Pacyna and Pacyna, 2001), which are mixed with mineral dust to varying degrees during atmospheric transport (Chester et al., 1993). EF values for V and Ni were > 10 in many non-NAF samples in the Mediterranean, consistent with shipping being a significant source of these elements (Becagli et al., 2012) but were also close to 1 in samples collected in the Black Sea, where shipping density is lower than in the Mediterranean. Although t-P was not notably enriched in NAF samples (EF values 1.6 – 6.2), we

have grouped it with the anthropogenic elements as many samples from other air mass types had EFs >10, suggesting that P (as well as V, Ni, Cu, Zn and Pb) has important non-natural dust sources (note that Mahowald et al. (2008) suggest that P enrichment might occur through the uplift of artificially fertilized soils).

Elemental ratios for the anthropogenic elements (Table S3) varied substantially between the different air masses encountered during GA04 as well as between reported values for the northeast Atlantic and eastern Mediterranean (Shelley et al., 2015; Herut et al., 2016). Relative to data collected in earlier decades (late 1970s to early 1990s), there has been a substantial decrease in concentrations and EF values for some elements (especially Pb, Cd) in both the Mediterranean and Black seas (Tables S4 & S5) due to the removal of Pb from petrol and other changes in anthropogenic emissions. However, EF values for Pb during GA04 were roughly an order of magnitude higher than values reported over the eastern tropical Atlantic in the early 2010s (1.8 – 15; Bridgestock et al., 2016) suggesting that the Mediterranean and Black Sea atmosphere were still subject to significant anthropogenic emissions of Pb in 2013.

### 3.5 Soluble anthropogenic elements (s-V, Ni, Cu, Zn, Cd, Pb) and phosphate

As with the lithogenic elements, concentrations of the soluble anthropogenic elements were lowest in the RNA samples during GA04, with most concentrations being below the limit of detection in that air mass type (Fig. 6). The profiles in Fig. 6 indicate relatively localised, independent sources for s-V, Ni, Cu, Zn, Cd, Pb and phosphate, although there are some common features between the elements. Most elements had relatively high concentrations in the northern Aegean Sea and Sea of Marmara (sample TM18). Soluble Cu, Cd and Pb had relatively high and increasing concentrations towards the end of the cruise near the Iberian Peninsula. Soluble (and total) V and Ni concentrations were strongly correlated ($r^2 = 0.973$ (soluble) and $r^2 = 0.955$ (total), both p = <<0.01). V and Ni in aerosols at Lampedusa in the central Mediterranean being primarily attributed to shipping emissions (Becagli et al., 2012), this is consistent with the idea that these emissions are the dominant factor controlling s-V and s-Ni concentrations, as previously observed in the Atlantic Ocean (Baker and Jickells, 2017).

Fractional solubilities for the anthropogenic elements were generally higher than those of the lithogenic elements, with most values being > 10 %. In contrast to the lithogenic elements, lowest fractional solubilities were not observed in the NAF samples even though EF values indicated that Saharan dust dominated the total element concentrations in these samples (Fig. 5). Phosphate was a possible exception to this with fractional solubilities of 6-36% in NAF samples, < 9% in RNA samples and a range of 14-49% in all other air mass types (Fig 6a).

### 3.6 Influences on (lithogenic) element solubility

During GA04, all the primarily lithogenic elements (Al, Ti, Mn, Fe, Co, Th) displayed inverse relationships between their total concentrations and fractional solubility (Fig. 7a, c, e, g, i & k). For some elements this is consistent with previously published studies. For instance, similar trends have been observed for Fe in a variety of global settings (Baker et al., 2006; Sholkovitz et al., 2009; Kumar and Sarin, 2010; Sholkovitz et al., 2012; Jickells et al., 2016; Shelley et al., 2018), including the eastern

Mediterranean (Theodosi et al., 2010a), and for Al and Ti in the Atlantic Ocean (Jickells et al., 2016; Shelley et al., 2018; Baker et al., 2020).

With the exception of Ti, the solubilities of the lithogenic elements during GA04 differ markedly from their previously reported behaviour over the Atlantic (see data from Jickells et al., 2016 plotted in Fig. 7). For Al and Fe, it is apparent that the NAF samples have generally lower fractional solubilities than for equivalent total element concentrations of Saharan dust aerosols over the Atlantic, while several of the more polluted (WEU and EEU) samples have higher solubilities than expected from the Atlantic total concentration– solubility relationship. The solubility of Mn in Saharan dust aerosols over the Atlantic has been reported to show little variability with atmospheric loading (Fig. 7e, Jickells et al., 2016; Shelley et al., 2018) and similar relationships have been reported for Co and Th in Saharan dust aerosols (Baker et al., 2020). During GA04, Mn, Co and Th solubility appeared to vary strongly with atmospheric loading. Solubilities for these elements in NAF samples, in particular, were substantially lower than previously observed in Saharan dust aerosols over the Atlantic at equivalent total element concentrations.

The elements designated as "anthropogenic" (P, V, Ni, Cu, Zn, Cd, Pb) did not generally show clear trends in fractional solubility with atmospheric loading and their solubilities were more consistent with previous results from the open Atlantic (Fig. 7b, d, f, h, j, l, & m; Jickells et al., 2016; Shelley et al., 2018). For most of these elements, solubilities in NAF samples were similar to those in other air mass types. Since the solubilities of anthropogenic sources of these elements are reported to be high at the point of emission (Desboeufs et al., 2005; Hsu et al., 2005; Li et al., 2022), there is presumably less potential for enhancement of solubility during atmospheric transport as exhibited by the lithogenic elements. Phosphate (P) is the exception to this, as it is the only element from this group to show higher solubility in the GA04 samples than in the Jickells et al. (2016) dataset from the Atlantic, for the dataset as a whole and specifically for the NAF / Saharan samples.

A number of mechanisms may potentially contribute to the enhancement of elemental fractional solubility during atmospheric transport (and solubility also varies between potential dust sources regions (Shi et al., 2011)). These mechanisms include chemical alteration by acidic species, change in mineral aerosol particle size (and hence surface area to volume ratio), change in mineralogy (also related to changes in particle size distribution) and photochemical redox change (see, for example, Baker and Croot, 2010). While we are lacking a number of pieces of information (aerosol particle size distributions for trace elements and acidic species, trace element redox states, trace element and acidic species mixing states, dust aerosol mineralogy) that might allow us to investigate these mechanisms in detail, the differences between the solubility behaviour observed during GA04 and that reported for aerosols over the Atlantic can be used to make inferences about the mechanisms of solubility control in the Mediterranean atmosphere.

Enhancement of trace element solubility in mineral dust at low pH has been demonstrated in the laboratory (Spokes and Jickells, 1996), inferred from observations of atmospheric aerosols (Fang et al., 2017) and incorporated into atmospheric chemistry models to improve simulations of soluble Fe deposition to the ocean (Myriokefalitakis et al., 2018). Acid species ($NO_3^-$ and nss-$SO_4^{2-}$) and (alkaline) $NH_4^+$ concentrations during GA04 were significantly higher (t test, $p < 0.05$) than in equivalent air mass types over the Atlantic Ocean (Jickells et al., 2016; Fig. 8). This is consistent with the more polluted nature

of the Mediterranean atmosphere and might be expected to lead to higher acid processing (or depression of acid processing, in the case of $NH_4^+$) of the NAF samples. However, the ratios of $nss\text{-}SO_4^{2-}$ / t-Fe in the NAF samples were lower than (but not significantly different to) these ratios in Saharan dust over the Atlantic and there does not appear to be an excess of acidic species to enhance acid processing in this case. Calcite content varies significantly with dust source (generally being higher in sources in the north of the Sahara (Chiapello et al., 1997; Kandler et al., 2007)) and may also impact solubility enhancement through neutralisation of acidity. However, while the $nss\text{-}Ca^{2+}$ / t-Fe ratio in the NAF samples (1.4-10.1 mol mol$^{-1}$) was relatively high, it was not outside the range of values for this ratio for Saharan dust in the Atlantic dataset (0.07-10.8 mol mol$^{-1}$). Variations in calcite content therefore seem unlikely to account for the observed differences between solubility over the Atlantic and Mediterranean. Enhancement of solubility through acid processing also requires that acidic species are internally mixed with mineral dust (i.e. both contained in the same aerosol particles, as opposed to external mixtures where particles of different composition occupy the same volume of air).

Ultimately, solubility enhancement through acid processing is dependent on the pH environment of the aerosol on an individual particle basis. This environment will vary strongly through the aerosol population, due to differences in internal mixing of acidic and alkaline species and trace elements (e.g. with particle size (Fang et al., 2017; Baker et al., 2020)). Furthermore, changes in the liquid water content of the particles (which is dependent on relative humidity and the hygroscopicity, and hence chemical composition, of the particles in question) can result in dramatic changes in pH, even when acid/alkaline ion balance varies little (Pye et al., 2020; Baker et al., 2021). Information about these factors is not available for the GA04 dataset and the insights provided by the above discussion of ion-solubility relationships are therefore limited.

Overall, the low solubility of Al, Mn, Fe, Co and Th in NAF samples (relative to similar samples collected over the Atlantic; Fig. 7) appears to be related to the short atmospheric transport pathway to the Mediterranean. This could lead to lower solubility through reduced internal mixing of acidic species with mineral dust particles (which together with slightly lower ratios of acidic species to lithogenic elements in NAF air masses would lead to lower atmospheric processing of dust particles), and a higher proportion of very large (low solubility) mineral particles relative to dust transported over the Atlantic Ocean. Short atmospheric transport times also reduce the potential for solubility of some elements to be enhanced through photochemical redox changes (e.g. insoluble Fe (III) to soluble Fe (II); Longo et al., 2016), although the very similar solubility behaviour of Fe and Al imply that redox changes are not a major control on Fe solubility (since Al has no redox chemistry). The enhanced solubility of Al, Mn, Fe, Co and Th in EEU and WEU samples (relative to equivalent total element concentrations over the Atlantic) may be due to a combination of both primary anthropogenic emissions of high solubility trace elements and secondary enhancement of solubility during transport to the Mediterranean and Black Sea basins due to atmospheric processing driven by pollutant acidic gases.

### 3.7 Dry deposition fluxes

Aerosol concentrations measured during GA04 have been used to estimate dry deposition fluxes for total inorganic nitrogen ($NO_3^-$ + $NH_4^+$), phosphate, mineral dust and soluble Fe, Mn, Ni, Zn and Cd (Table 2). Dry deposition fluxes for the other

soluble trace elements are given in Table S6. This approach provides a snapshot of the dry deposition flux that may not be representative of fluxes over longer timescales. However, our estimates appear broadly comparable with fluxes reported from longer-term sampling around the Mediterranean when annual averages are expressed on a daily basis (e.g. DIN 70-126 µmol m$^{-2}$ d$^{-1}$ (western basin) and 79-210 µmol m$^{-2}$ d$^{-1}$ (eastern basin), (both Markaki et al., 2010) and 166 µmol m$^{-2}$ d$^{-1}$ (Crete) (Theodosi et al., 2019); dust 35 mg m$^{-2}$ d$^{-1}$ (Crete) (Theodosi et al., 2019) and 3.8-5.3 mg m$^{-2}$ d$^{-1}$ (Corsica) (Desboeufs et al., 2018). Our estimate of t-Al (dust) deposition to the Black Sea is also similar to the value reported by Theodosi et al. (2013) based on sampling at two coastal sites (~ 6 mg dust m$^{-2}$ d$^{-1}$, assuming Al is 8 % of dust by mass). Rainfall rates in the Mediterranean and Black Sea basins during summer are low: 0.72 mm d$^{-1}$ for the western Mediterranean (west of Sicily), 0.36 mm d$^{-1}$ for the eastern Mediterranean, and 1.88 mm d$^{-1}$ for the Black Sea (Adler et al., 2003). Dry deposition therefore probably accounts for the majority of total atmospheric fluxes to the basins over the study period, although rainfall composition data over the open Mediterranean are extremely scarce (Desboeufs et al., 2022) so this is difficult to verify.

Note that the low EF values observed for some "anthropogenic" elements (e.g. P, V, Ni, Cu and Zn for some samples during Leg 1 in the eastern Mediterranean, and V and Ni during Leg 2; Fig. 5) may imply a higher proportion of the soluble fractions of these elements in coarse aerosols than in other samples encountered during GA04. This, in turn, may suggest that deposition velocities in these cases might be higher than the value (0.1 cm s$^{-1}$) used in our calculations. Thus, deposition fluxes in these cases may be higher than those given in Table 2, although the absence of aerosol size distribution data for these soluble elements makes the magnitude of such underestimation difficult to quantify.

Table 2. Mean, median and ranges of dry deposition fluxes for total inorganic nitrogen (TIN), phosphate (PO$_4$), mineral dust, soluble Fe (s-Fe), Mn (s-Mn), Ni (s-Ni), Zn (s-Zn) and Cd (s-Cd) over the western and eastern Mediterranean and Black Sea basins. Note that no data (ND) was collected for TIN or PO$_4$ during the Black Sea leg of GA04.

| | TIN | PO$_4$ | dust | s-Fe | s-Mn | s-Ni | s-Zn | s-Cd |
|---|---|---|---|---|---|---|---|---|
| | µmol m$^{-2}$ d$^{-1}$ | nmol m$^{-2}$ d$^{-1}$ | mg m$^{-2}$ d$^{-1}$ | nmol m$^{-2}$ d$^{-1}$ | nmol m$^{-2}$ d$^{-1}$ | nmol m$^{-2}$ d$^{-1}$ | nmol m$^{-2}$ d$^{-1}$ | nmol m$^{-2}$ d$^{-1}$ |
| **Western Med** | | | | | | | | |
| Mean | 190 | 52.7 | 4.1 | 347 | 109 | 6.10 | 19.9 | 0.087 |
| Median | 148 | 47.6 | 3.4 | 248 | 73.3 | 3.60 | 12.9 | 0.080 |
| min | 95.1 | 16.3 | 0.33 | 10.9 | 1.40 | 0.05 | 3.98 | 0.005 |
| max | 398 | 132 | 12.9 | 1160 | 395 | 44.2 | 73.0 | 0.241 |
| **Eastern Med** | | | | | | | | |
| Mean | 164 | 73.4 | 21.2 | 224 | 140 | 1.99 | 32.2 | 0.099 |
| Median | 121 | 70.1 | 6.5 | 122 | 52.9 | 1.02 | 21.0 | 0.075 |
| min | 90.7 | 17.8 | 1.7 | 10.4 | 0.8 | 0.03 | 2.2 | 0.009 |
| max | 382 | 189 | 111 | 1200 | 1087 | 9.58 | 118 | 0.270 |
| **Black Sea** | | | | | | | | |
| Mean | ND | ND | 8.6 | 217 | 198 | 2.69 | 23.1 | 0.086 |
| Median | ND | ND | 7.3 | 156 | 71 | 2.87 | 15.7 | 0.085 |
| min | ND | ND | 3.0 | 97 | 25 | 0.33 | 6.07 | 0.019 |
| max | ND | ND | 14.2 | 705 | 950 | 6.33 | 43.3 | 0.224 |

### 3.7.1 Mediterranean Sea

Total inorganic nitrogen and phosphate dry deposition fluxes were of the same order for the western and eastern Mediterranean basins. However, the ratio of N/P was higher in the western basin (median = ~3100 and 1700 mol mol$^{-1}$ in the western and eastern basins, respectively). This is consistent with the regional gradient in N/P in annual and seasonal atmospheric deposition derived from modelling studies (Okin et al., 2011; Kanakidou et al., 2020). These N/P ratios suggest that atmospheric deposition contributes to the phosphate-limited status of the eastern Mediterranean (Krom et al., 2010; Markaki et al., 2010).

Previous observations of atmospheric deposition over an annual cycle have shown total Fe (which is dominated by mineral dust) flux to be higher in the western than in the eastern Mediterranean (Guieu et al., 2010b). However, during GA04, the highest (by an order of magnitude) calculated mineral dust fluxes were in the eastern basin (Table 2). Dust transport to the Mediterranean is highly seasonal, with peak activity starting in the late spring / early summer in the eastern basin before migrating into the western basin later in the summer (Moulin et al., 1998; Querol et al., 2009). The GA04 dust fluxes were broadly consistent with these patterns of dust transport, especially when the more northerly (and remote from dust sources) route of Leg 3 in the western basin, the short timescale of the GA04 sampling and the sporadic nature of dust transport events (e.g. Guieu et al., 2010a; Guieu et al., 2010b) are taken into account.

The much higher maximum deposition in the eastern basin observed for dust was not apparent for the soluble elements (Fe, Mn, Ni, Zn and Cd; Table 2). This was despite the dominance of dust as a source of Fe and Mn (EFs < 2). Thus the higher fluxes of s-Fe and s-Mn relative to dust in the western basin probably reflect the prevalence of WEU airmasses encountered there during GA04 and the higher fractional solubility of both elements in this airmass type. The west / east distribution of s-Ni, Zn and Cd fluxes also appears to be impacted by the air mass regimes encountered during sampling. All these elements have high fractional solubilities that do not vary strongly between air mass types, so their higher soluble fluxes in the west are due directly to the higher prevalence of polluted air (average EFs in the western basin were at least double those in the east for all these elements). The EF values observed for GA04 are in broad agreement with the results of Guieu et al. (2010b) who estimated that ~ 89% of the Fe deposition flux to the Mediterranean basin was of lithogenic origin, whereas Zn and Cd were predominantly of anthropogenic origin (~ 88 and 96%, respectively).

Middag et al. (2022) used a simple water balance model approach combined with water column measurements made during GA04 to suggest that atmospheric inputs of Mn, Ni, Zn and Cd were required to balance the Mediterranean budgets of these elements. Assuming that the aerosol samples collected during GA04 were representative of deposition to the region, mean dry deposition fluxes (Table 2) during summer (June - August) can account for 11 (3.6 – 32) % (Mn), 3.2 (1.1 – 9.6) % (Ni), 8.6 (2.9 – 26) % (Zn) and 1.0 (0.3 – 2.9) % (Cd) of the annual deficit in the surface budgets reported by Middag et al. (2022) in the western Mediterranean and 1.4 (0.5 – 4.3) % of the Ni deficit in the eastern basin (values in parentheses represent the range due to a 3-fold uncertainty in deposition velocity). The values for Ni in the eastern basin may be lower limits, as noted above. Middag et al. did not report a deficit of Mn, Zn or Cd in the eastern Mediterranean.

### 3.7.2 Black Sea

Calculated dry deposition fluxes of dust in the Black Sea were similar to those in the western Mediterranean during GA04 (Table 2). Arrivals from North Africa have been reported to account for 10% of air mass arrivals over the Black Sea in summer (Kubilay et al., 1995), although similar arrivals were not observed during GA04. However, regions of Europe to the northwest of the Black Sea (which dominated air mass arrivals during Leg 2 of GA04) have previously been reported to be significant sources of crustal elements to the basin (Hacisalihoglu et al., 1992). Modelling studies indicate that the annual average t-Fe (and dust) atmospheric flux to the Black Sea is ~2 orders of magnitude lower than to the Mediterranean (Myriokefalitakis et al., 2018), but the short-term sampling conducted during GA04 do not represent these longer-term trends.

Soluble-Fe and Mn dry deposition fluxes to the Black Sea were similar to those to the Mediterranean. As for the western Mediterranean, these elements appeared to have predominantly crustal sources (Fig. 3c & d). However, their relatively high fractional solubility over the Black Sea (Fig. 4 c & d) suggests a higher influence for anthropogenic emissions than observed in the eastern Mediterranean. Unusually for the elements studied here, Ni also followed this pattern of low EF (Fig. 5c) and high fractional solubility (Fig. 6c) in the Black Sea. Soluble Zn and Cd dry deposition fluxes in the Black Sea were unequivocally associated with anthropogenic activities (Fig. 5e & f).

## 4 Conclusions

Aerosol sampling during GA04 has confirmed that trace elements over the Mediterranean and Black seas were influenced by mineral dust and various anthropogenic sources from western and eastern Europe. Although the lithogenic elements (Al, Ti, Mn, Fe, Co, Th) are dominated by crustal sources, their fractional solubilities vary considerably with atmospheric dust load. In most cases, lithogenic element solubilities showed more intense responses to changing dust load than observed in the atmosphere over the Atlantic Ocean, with the GA04 samples having lower solubility in dusty aerosols and often higher solubility in polluted European air masses. The shorter atmospheric transport (and hence lower atmospheric processing and less significant mixing with anthropogenic emissions) of Saharan dust to the Mediterranean than to the Atlantic is probably a major contributor to this behaviour. The distributions of the highly soluble anthropogenic elements (P, V, Ni, Cu, Zn, Cd, Pb) indicated that they had a variety of diverse sources around the European continent, as well as from intense shipping activity in the Mediterranean. Our calculated dry deposition flux values confirm the gradient in the N/P ratio of atmospheric deposition across the Mediterranean basin and that atmospheric deposition can be a significant source of soluble elements (especially Mn and Zn) to the basin.

**Data availability:**

The data reported in this manuscript have been submitted to the GEOTRACES IDP (soluble element data were submitted to IDP2017, other data is currently being submitted for inclusion in IDP2025) and are also available from the corresponding author on request.

**Author contribution:**

Writing – original draft preparation: RUS, Writing – review & editing: All authors, Investigation: MT, SM, RUS, Formal analysis: All authors, Conceptualization and Funding acquisition: ARB.

**Competing interests:**

The authors declare that they have no conflict of interest.

**Acknowledgements**

We gratefully acknowledge the support and assistance of Micha Rijkenberg, the chief scientist of GA04, Pim Boute (Leg 1), Morten Andersen (Leg 2), Eyal Wurgaft and Simona Brogi (Leg 3) for sampling, and the captain and crew of RV *Pelagia* during the voyage. This work was funded by the Dutch Research Council (project number 822.01.015), the UK Natural Environment Research Council (grant number NE/V001213/1) and the University of East Anglia. The International GEOTRACES Programme is possible in part thanks to the support from the U.S. National Science Foundation (OCE-2140395) to the Scientific Committee on Oceanic Research (SCOR). Global Precipitation Climatology Project (GPCP) Monthly Analysis Product data was provided by the NOAA PSL, Boulder, Colorado, USA, from their website at https://psl.noaa.gov. We thank Karine Desboeufs and two anonymous reviewers for their helpful comments.

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

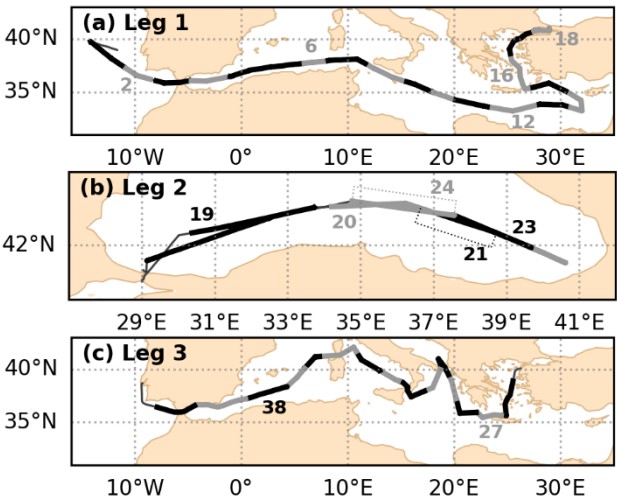

Figure 1. GA04 cruise track showing aerosol sampling periods as alternating thick black and grey bars for (a) Leg 1, (b) Leg 2 and (c) Leg 3. Note that the eastbound and westbound sections of Leg 2 followed very similar tracks and cannot be distinguished on this map. Numbers for samples referred to in the text and in Fig. 2 are indicated.

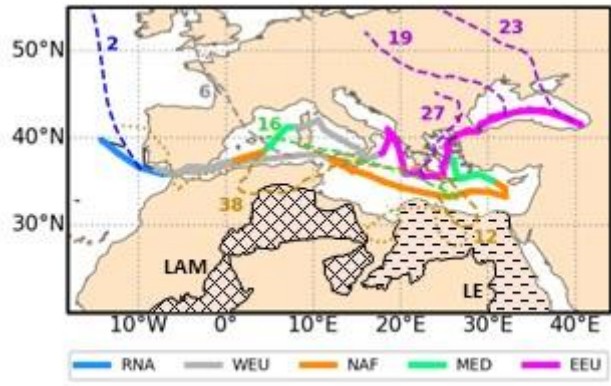

Figure 2. GA04 cruise track showing the classification of aerosol samples into 5 air mass types. Example AMBTs (and sample numbers) for each type are shown for arrivals at 10 m (dashed lines) and (for NAF only) 1000 m (dotted lines). The Libya-Algeria-Mali (LAM) and Libya-Egypt (LE) Potential Source Areas (Guinoiseau et al., 2022) are also shown.

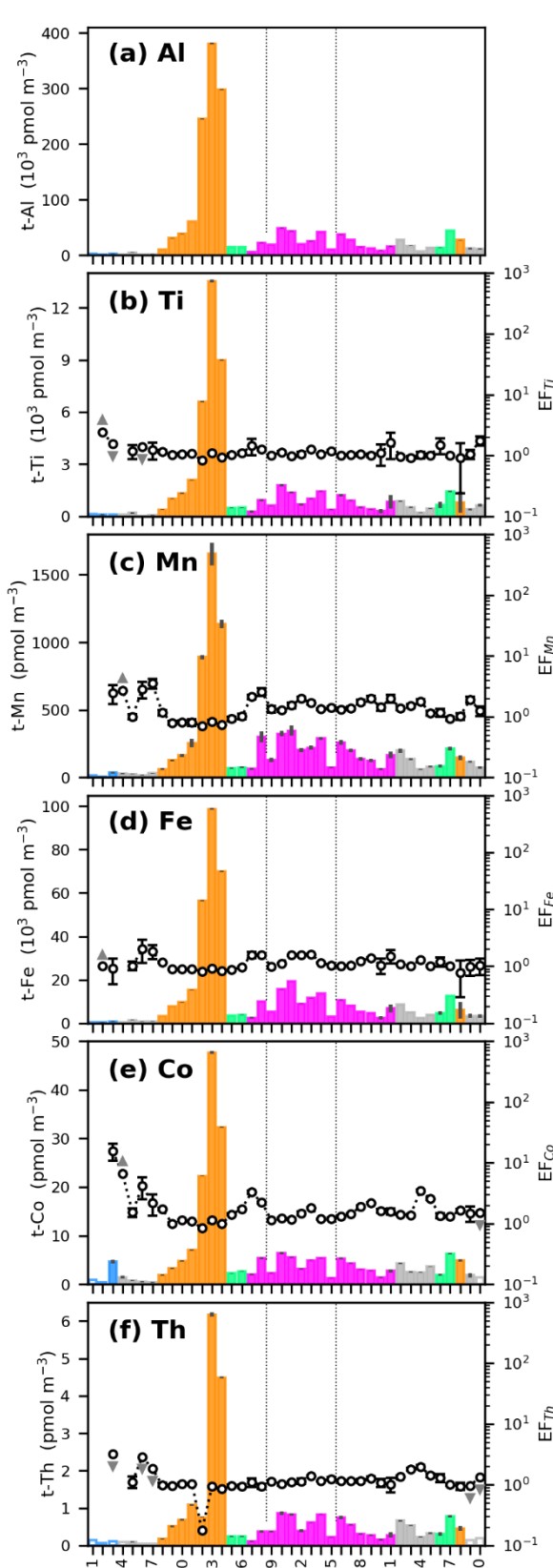


**Figure 3.** Total concentrations of lithogenic elements ± 1 SD (pmol m$^{-3}$), with enrichment factors relative to Al overlaid (circles). Unfilled bars indicate that analyte was below the limit of detection and bar represents 75% of the limit of detection. An EF was not determined if both Al and the element of interest were below the limit of detection. Up- / down-ward pointing grey arrows near EF markers indicate that values are minima / maxima because Al / the element were below the limit of detection. Bars are coloured according to the air mass type of each sample, blue = RNA, grey = WEU, orange = NAF, green = MED, pink = EEU. The dashed grey vertical lines indicate the legs of the cruise, with Leg 1-3 being left to right.


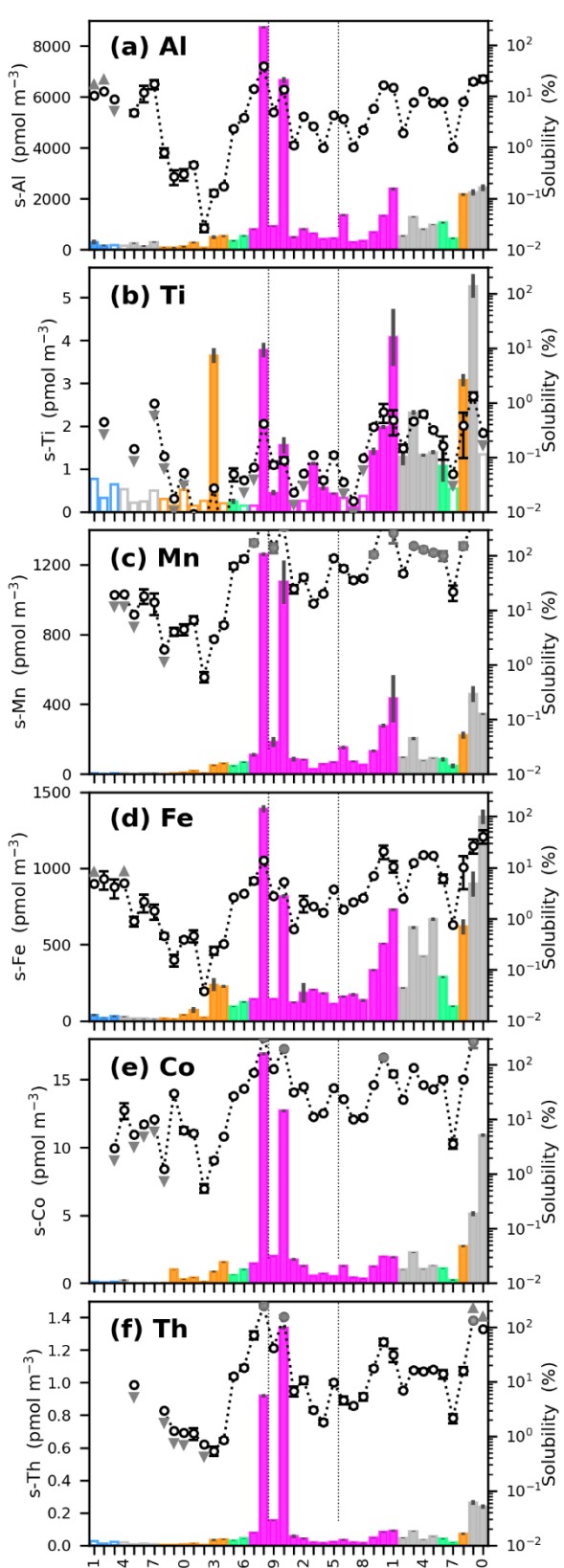

**Figure 4. Soluble concentrations of lithogenic metals ± 1 SD (pmol m$^{-3}$), with fractional solubility overlaid (circles). Up- / down-ward pointing grey arrows near solubility markers indicate that values are minima / maxima because total / soluble concentrations were below the limit of detection. Note that the axis for fractional solubility is capped at 300%. This is because there were a few instances of solubility >100% (grey circles), suggesting some low-level contamination of those samples. Rather than set these solubilities to a maximum of 100%, we close this approach to make it clear which samples this applied to. Bar colours and dashed vertical lines are described in Fig. 3.**

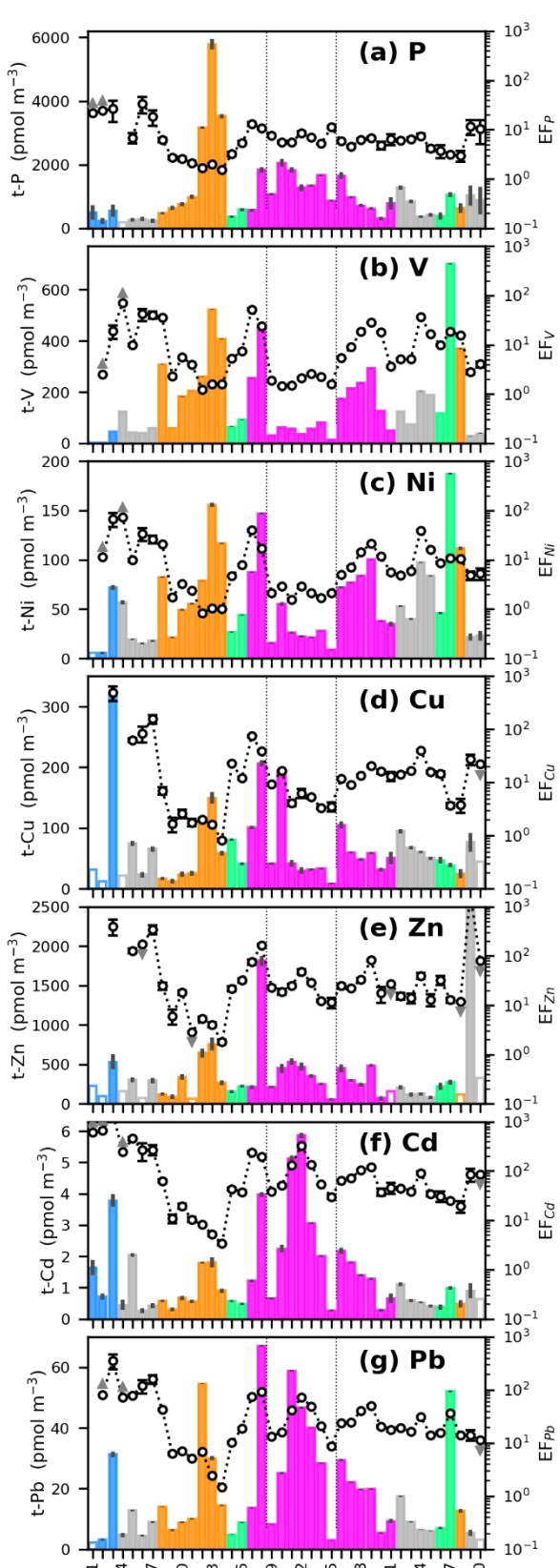


**Figure 5. Total concentrations of anthropogenic elements ± 1 SD (pmol m$^{-3}$), with enrichment factors overlaid. Note that Zn contamination of sample TM39 (10500 pmol m$^{-3}$) is strongly suspected so the y-axis for the Zn data is limited to 2500 pmol m$^{-3}$. Similarly, the EF$_{Zn}$ axis is limited to 1000 as the calculation for EF$_{Zn}$ for TM39 was made using the suspect data point. Bar colours, grey arrows and dashed vertical lines are described in Fig. 3.**


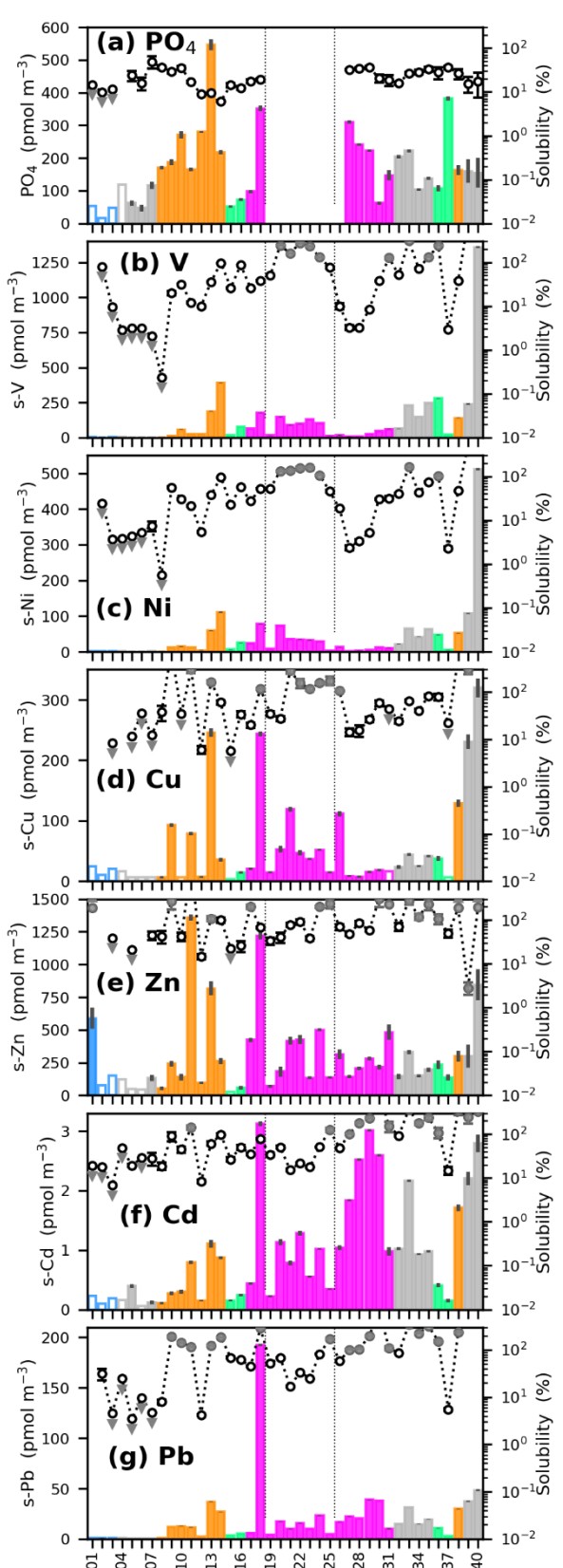

**Figure 6. Soluble concentrations of anthropogenic elements and phosphate $\pm$ 1 SD (pmol m$^{-3}$), with fractional solubility overlaid. There is no phosphate data from Leg 2 (Black Sea) as the sampler malfunctioned on this leg. Bar colours, grey circles and arrows and dashed vertical lines are described in Figs. 3 & 4. Note that Zn solubility for sample TM39 is affected by contamination of the t-Zn measurement.**

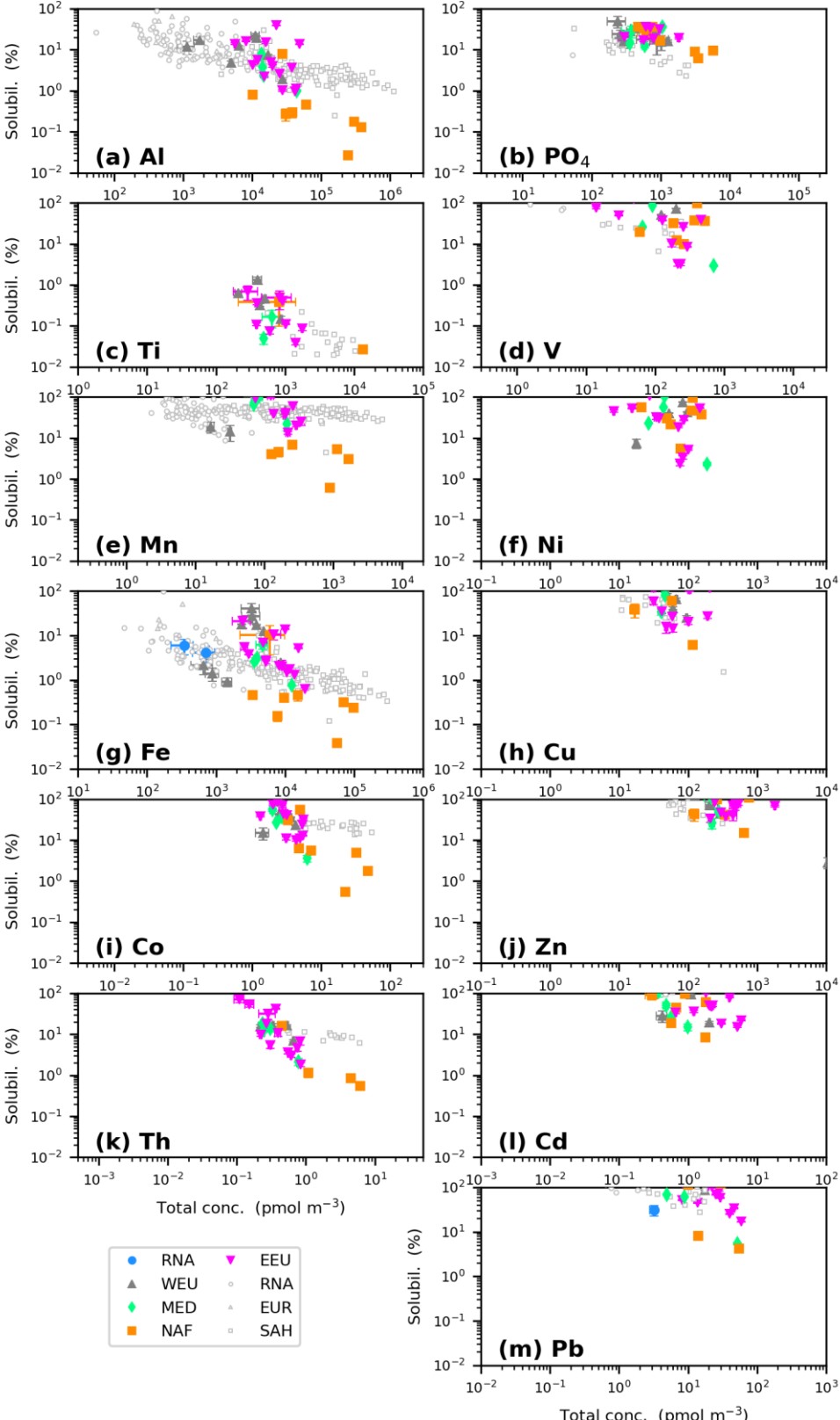

**Figure 7. Log-log plots of fractional solubility plotted against total aerosol concentration (pmol m⁻³) for primarily lithogenic (left column) and anthropogenic (right column) elements. Samples are colour coded by air mass, as in previous figures. The grey open symbols show data for RNA, European (EUR) and Saharan (SAH) air mass types over the open Atlantic Ocean from Jickells et al. (2016) for comparison.**

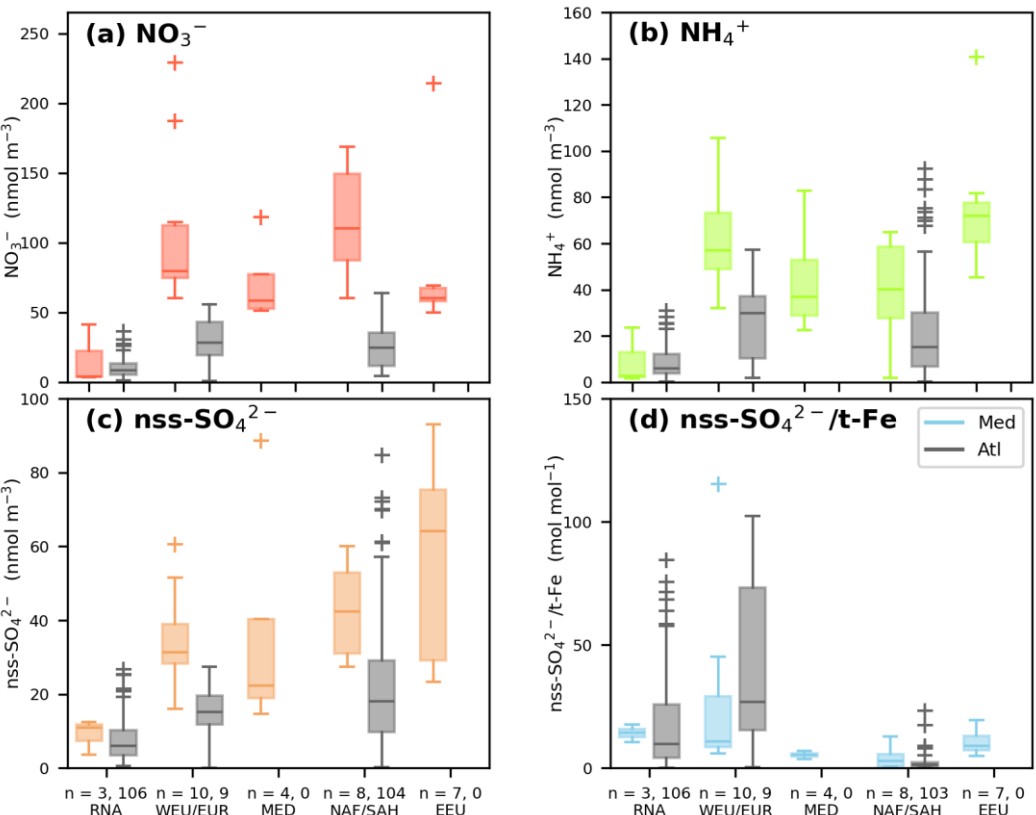

**Figure 8. Box and whisker plots of aerosol concentrations of (a) NO₃⁻, (b) NH₄⁺, (c) nss-SO₄²⁻ (nmol m⁻³) and (d) the molar ratio of nss-SO₄²⁻ to total-Fe from GA04 (coloured boxes) and the Atlantic Ocean (grey boxes; Jickells et al., 2016). Note that WEU and NAF used in this study correspond with EUR and SAH, respectively, from earlier studies from this research group. Crosses indicate outlier values greater than 1.5 times the interquartile range above the upper quartile.**