# Peer review of "Aerosol trace element solubility and deposition fluxes over the Mediterranean and Black Sea basins"

_EGUsphere, 2024_

## Referee Comment (RC2)

**Review of "Aerosol trace element solubility and deposition fluxes over the polluted, dusty Mediterranean and Black Sea basins" by R. Shelley et al.**

Shelley et al. present a novel dataset of trace metals and P solubility and dry atmospheric fluxes from aerosols collected during GA04 cruise in Mediterranean (Med) and Black seas. It is interesting to have atmospheric data useful for studies on air-sea exchange in these regions which are influenced by various aerosol sources, notably with a gradient between East and West,.

The manuscript is well-written, logically organized, and adequately illustrated. Abstract is succinct and accurate. The discussion focuses in part on the comparison between the values found in the Med and Black Seas compared with the ones obtained in Atlantic Ocean, where the authors took part in several cruises. This comparison is interesting in order to situate the anthropogenic contribution in these two regions, which are not at the same distance from pollution sources. It would also have been useful to have a discussion in this context with the other measurements made in the Mediterranean, in particular for deposition fluxes. There is extremely little data available from campaigns at sea, but there are several observatories on islands with long-term measurements in these regions, e.g. Finokalia, Lampedusa or data from annual monitoring: Corsica, Balearic island.... A small putting in context and comparison with previous and recent works (since the date have been obtained in 2013) could be unvaluable to get the most out of these data.

Overall, the quality of the figures is not very good, but I think this is due to the changeover to pdf format and should be sorted out in the final manuscript.

**Specific comments:**

**Title:** Because of their location, the Mediterranean and the Black Sea are usually subject to polluted or desert dust-laden masses. In the case of this paper, no specific or intense events of desert dust or pollution have been reported. So I think the title could be simplified: "Aerosol trace element solubility and deposition fluxes over the Mediterranean and Black Sea basins"

**Figure 1**: In order to have a better view of the transect and the sampling zones, it would be useful to zoom in on the measurement area over sea (e.g., between 50°N and 25°N) and refine the thickness of the aerosol track.

It would be also useful to include the numbers of the filters that are noteworthy and discuss them in the text, for example, samples 18 and 20, which stand out in terms of soluble concentrations.

**SC-01**. **Line 95-134**. The description of the field blanks (sampling and analysis) is clear, but there is no clear explanation regarding the use of the blank values. Are they subtracted from the total and soluble values? If so, which 'type' of blanks (1, 2, or 3)?

**SC-02: Line 50:** "*For our flux calculations, we used $v_d$ values of 1 cm s$^{-1}$ for the lithogenic elements and 0.1 cm s$^{-1}$ for the anthropogenic elements*". The data show that certain elements are predominantly crustal during desert dust events (see EF values), whereas they are anthropogenic at other times. It would be interesting to calculate the percentage of desert dust inputs compared to anthropogenic inputs during the campaign. If it is significant, in order to better constrain the flux calculations, shouldn't these dust events (i.e. vd values of lithogenic) be taken into account in the flux calculations for these elements?

**SC-03: 3.1. Air mass origins during GA04 :**

The analysis of back-trajectories is useful for classifying the different samples based on potential sources, but a more detailed analysis of the actual sources might help better understand the various influences in the discussion section. To aid data analysis in the case of desert dust arrivals, it would be interesting to examine dust emissions, using observations such as RGB SEVIRI, along the back-trajectories, rather than relying on PSAs which do not provide information on the actual origin of the transported dust. Indeed, just because an air mass passes over potentially emitting areas does not mean it picks up desert dust; the winds must be strong enough to trigger dust production. For example, in the case of sample 38, the air mass passes over both Tunisia and Algeria, which are two emissive sources with quite different mineralogical characteristics. The Tunisian source is richer in calcite, which can play a role in neutralizing acids (Kandler et al., 2007). It coud be invaluable to consider this point in the discussion of acid effect on solubility.

*Kandler, K., Benker, N., Bundke, U., Cuevas, E., Ebert, M. and co-authors. 2007. Chemical composition and complex refractive index of Saharan Mineral Dust at Izaña, Tenerife (Spain) derived by electron microscopy. Atmos. Environ. 41, 8058–8074.*

**Figure 2:** This figure is complex enough. To clarify Figure 2, perhaps using only the back-trajectories at 10 meters might be sufficient since the collection takes place at the height of the ship? Furthermore, the text does not explain which altitude (10, 500, or 1000 meters) is used as the reference for classifying the air masses. It would be useful to specify, if possible, the altitudes of the boundary layer during the campaign, as the back-trajectories at 1000 meters are only relevant if the boundary layer reaches at least 1 km and is well mixed during the sampling?

Could you add the filter numbers at the beginning and end of the back-trajectory types to Figure 2 to link this figure with the subsequent ones, especially in the area where the routes of legs 1 and 3 overlap?

In the same idea, please color the bars in Figures 3, 4, 5, and 6 according to the air mass origins (rather than just the colored horizontal bar at the top of each panel). This would make it much easier for the reader to link the sample to its origin (which is the basis of your data analysis in the following sections). Since the names of the metals are labeled on each panel, it should not cause any confusion.

**SC-04 : Lithogenic elements**

**L180** : Please indicate in the text the average value of metal concentrations (e.g., iron) in the NAF and non-NAF samples, so that these values can be used as a reference for metal concentrations according to their origin, and to quantify the contribution of Saharan air masses to the background metal levels during the campaign

**L200 and 201** : I don't understand the connection between the first sentence and the given example since s-Fe concentrations in the RNA samples are lower than in the NAF samples.

**L200 to 205** : In addition to the s-Fe values, include the average solubility value. This will help illustrate your discussion on the relationship between solubility and s-Fe concentrations.

**SC-05 : Anthropogenic elements**

**L219** : add ", except P" before Fig.5

**L223 then 242**: *"Soluble V and Ni concentrations were strongly correlated (r2 = 0.973, p = <<0.01),*

*which is consistent with V and Ni in aerosols being markers of shipping emissions, and these emissions being the dominant control on s-V and s-Ni concentrations (Becagli et al., 2012; Baker and Jickells, 2017)."*

Here, I think that it will be more precise to write: "Soluble V and Ni concentrations were strongly correlated ($r^2$ = 0.973, p << 0.01). V and Ni in aerosols being primarily attributed to shipping emissions (Becagli et al., 2012), this is consistent with the idea that these emissions are the dominant factor controlling s-V and s-Ni concentrations, as previously observed in the Atlantic Ocean (Baker and Jickells, 2017)."

In order to reinforce this conclusion, is there a correlation between the t-Ni and t-V values in the non-NAF samples that could confirm a source related to maritime traffic, as observed by Becagli et al. (2012)? It could be interesting to confirm (or not) the Becagli et al.'s conclusion obtained on local point in Med, on a large area of Med and Black seas. If it is not the case, the fact that s-Ni and s-V are correlated indicate that even if these metals can have different anthropogenic sources (that is highly possible in Med and consistent with your sentence L223-225), the 'shipping' source, is the primary contributor of soluble atmospheric inputs for these metals.

**L 225**: "*suggesting that P (as well as V, Ni, Cu, Zn and Pb) has important non-dust sources*": P could be issued from European dust source related to agricultural soils enriched in P by fertilizers. That could explain that t-P follows the behavior of lithogenic elements, even if the EF are >10 (see e.g. Bergametti data in Mahowald et al. (2008)).

*Mahowald, N., et al. (2008), Global distribution of atmospheric phosphorus sources, concentrations and deposition rates, and anthropogenic impacts, Global Biogeochem. Cycles, 22, GB4026, doi:10.1029/2008GB003240*

**L233-235** : It would be more correct to write : «… *that the Mediterranean and Black Sea atmospheres were still subject to significant anthropogenic influence of Pb emissions sources in 2013*", since these data are more than 10 years old (European anthropogenic Pb emissions decreased since the cruise, see e.g. annual European Union emission inventory report 1990-2022) and the emissions come from the proximity of various anthropogenic sources compared with the Atlantic.

**Figure 4** : Authors explained that some values of solubility reach 300% due to probable contaminations. I am rather sceptical about the presentation of these values in the same way as the other values, since it's impossible to estimate the contamination and these values suggest that the solubility is high when it may only be a few %, for example. I think it would be clearer to show these values in "transparent" colour, rather than in the same colour as the uncontaminated values.

**SC-06 : Influence on element solubility**

**L298-309**: Your discussion is based on the acid effect from polluted species, but your dissolution protocol is also carried out in a buffered medium. To what extent could the excess acid affect the values you find?

Another point that raises questions in this part of the discussion, although the concentrations of inorganic acid and base species are higher than in the Atlantic, it is the capacity of these species to acidify or neutralise the aerosol, and therefore play on the pH, that will affect solubility. For this, it is not so much the concentrations that are important, but the balance of neutralisation of the inorganic (and organic) acidic species by the alkaline species (or vice versa). You use the nssSO4/Fe ratio to estimate the excess of acid over Fe, but this is only valid if nssSO4 is not neutralised by $NH_4^+$ or by the presence of calcite, for example, in your samples. In your case, as you do not have Ca values, you cannot estimate the neutralisation capacity linked to the presence of calcite in NAF samples, but you

can estimate the effect of $NH_4^+$ in all your samples. It would be more accurate to use, for example, the neutralisation ratio (Silvern et al., 2017) rather than the nssSO4/Fe value.

*Silvern, R. F., Jacob, D. J., Kim, P. S., Marais, E. A., Turner, J. R., Campuzano-Jost, P., and Jimenez, J. L.: Inconsistency of ammonium–sulfate aerosol ratios with thermodynamic models in the eastern US: a possible role of organic aerosol, Atmos. Chem. Phys., 17, 5107–5118, https://doi.org/10.5194/acp-17-5107-2017, 2017.*

**L314** : *"Overall, the low solubility of Al, Mn, Fe, Co and Th in NAF samples (relative to similar samples collected over the Atlantic; Fig. 7) appears to be related to the short atmospheric transport pathway to the Mediterranean."* : The low solubility 'could be' related, this is one possible explanations but it would also be interesting to discuss the effect of other phenomena to explain this low solubility compared to the values in the Atlantic. First of all, the dust emitting sources are different between the samples taken in the Mediterranean (North of Africa) and those taken in the Atlantic (Morocco, Central Sahara, Bodélé, Sahel, etc.). The difference in sources (including the difference in mineralogy, particle size distribution and calcite load, and therefore pH) could also explain the differences you observe. You seem to consider that all Saharan emission sources have the same solubility (and should respond in the same way to the process along the transport route), but the data at the source show that the solubility of iron in particular can vary from one source to another (see e.g. Shi et al., 2011; Paris et al., 2010). In particular, the solubility in the first dust period appears to be lower than in the second NAF period, even though the route is comparable.

Moreover, unless I'm mistaken, most of the measurements made in the Atlantic were taken during a different period than the GA04 cruise and the seasonality of the sources is different. It would be interesting to specify this in the text, particularly as photochemical processes are probably accentuated in summer in the Med and the Black Sea. This could perhaps explain the differences in solubility in the case of non-NAF samples, e.g. linked to oxidation processes or to the greater presence of organic matter or potential mixing with anthropogenic forms.

*Shi, Z., et al. (2011), Influence of chemical weathering and aging of iron oxides on the potential iron solubility of Saharan dust during simulated atmospheric processing, Global Biogeochem. Cycles, 25, GB2010, doi:10.1029/2010GB003837*

*Paris, R., Desboeufs, K. V., Formenti, P., Nava, S., and Chou, C.: Chemical characterisation of iron in dust and biomass burning aerosols during AMMA-SOP0/DABEX: implication for iron solubility, Atmos. Chem. Phys., 10, 4273–4282, https://doi.org/10.5194/acp-10-4273-2010, 2010.*

**SC–07 Dry deposition**

**L 331: "Dry deposition therefore probably accounts for the majority of total atmospheric fluxes to the basins over the study period."** To reach this conclusion, rather than compare with the rainfall, it could be interesting to compare with the elemental fluxes obtained by wet or total deposition measurements in the same period in these areas. For example, if we consider an iron concentration of 340 $nmol.L^{-1}$ in rain collected in June in the western Mediterranean (Desboeufs et al., 2022, it is just an example as base for the calculation ) with a rainfall of 1 $mm.d^{-1}$, this is equivalent to a flux of 340 $nmol.m^{-2}.d^{-1}$, i.e. in the average of values obtained here.

*Desboeufs, K., et al.: Wet deposition in the remote western and central Mediterranean as a source of trace metals to surface seawater, Atmos. Chem. Phys., 22, 2309–2332, https://doi.org/10.5194/acp-22-2309-2022, 2022.*

Only Guieu's work is cited, even though it dates from the early 2000s and several studies on deposition have since been carried out in these regions. Numerous measurements have been made in the Mediterranean and there is a great deal of data available (e.g. the long-term measurements in Finokalia, the literature on the total deposition in Sardinia, Corsica or Balearics Islands ). It is a pity that no comparison is provided with these data obtained on islands to see the potential differences with our data, notably about the source of trace metals and nutrients, which are studied in this literature, e.g.:

*Theodosi C., Markaki Z., Pantazoglou F., Tselepides A., Mihalopoulos N., Chemical composition of downward fluxes in the Cretan Sea (Eastern Mediterranean) and possible link to atmospheric deposition: A 7 year survey, Deep-Sea Research Part II, 164, 89-99, 2019.*

*Desboeufs, K., Bon Nguyen, E., Chevaillier, S., Triquet, S., and Dulac, F.: Fluxes and sources of nutrient and trace metal atmospheric deposition in the northwestern Mediterranean, Atmos. Chem. Phys., 18, 14477–14492, doi.org/10.5194/acp-18-14477-2018, 2018.*

*Christodoulaki S., G. Petihakis, N. Mihalopoulos, K. Tsiaras, G. Triantafyllou, M. Kanakidou, Human-Driven Atmospheric Deposition of N and P Ccontrols on the East Mediterranean Marine Ecosystem, JAS, 73, 1611- 1619, 2016.*

*Kanakidou M., S. Myriokefalitakis, N. Daskalakis, G. Fanourgakis, A. Nenes, A. Baker, K. Tsigaridis, N. Mihalopoulos, Past, Present and Future Atmospheric Nitrogen Deposition, JAS, 73, 2039-2047, 2016.*

*Longo, A. F., Ingall, E. D., Diaz, J. M., Oakes, M., King, L. E., Nenes, A., Mihalopoulos, N., Violaki, K., Avila, A., Benitez-Nelson, C. R., Brandes, J., McNulty, I., and Vine, D. J.: P-NEXFS analysis of aerosol phosphorus delivered to the Mediterranean Sea, Geophys. Res. Lett., 41, 4043–4049, https://doi.org/10.1002/2014GL060555, 2014.*

*Im U., S. Christodoulaki, K. Violaki, P. Zarbas, M. Kocak, N. Daskalakis, N. Mihalopoulos and M. Kanakidou, Atmospheric deposition of nitrogen and sulfur over Europe with focus on the Mediterranean and the Black Sea, Atmospheric Environment, 81, 660-670, 2013.*

*Markaki Z., M.D. Loye-Pilot, K. Violaki, L. Benyahya, N. Mihalopoulos, Variability of atmospheric deposition of dissolved nitrogen and phosphorus in the Mediterranean and possible link to the anomalous seawater N/P ratio, Marine Chemistry, Volume 120, Issues 1-4, Pages 187-194, 2010.*

*Theodosi C., Z. Markaki, A. Tselepides, N. Mihalopoulos, The significance of atmospheric inputs of soluble and particulate major and trace metals to the eastern Mediterranean seawater, Marine Chemistry, Volume 120, Issues 1-4, 20, 154-163, 2010.*

*Theodosi C., Z. Markaki, N. Mihalopoulos, Iron speciation, solubility and temporal variability in wet and dry deposition in the Eastern Mediterranean, Marine Chemistry, Volume 120, Issues 1-4, 20, 100-107, 2010.*

*Guerzoni, S., Molinaroli, E., Rossini, P., Rampazzo, G., Quarantotto, G., and Cristini, S.: Role of desert aerosol in metal fluxes in the Mediterranean area, Chemosphere, 39, 229–246, https://doi.org/10.1016/S0045-6535(99)00105-8, 1999.*

*Frau, F., Caboi, R., and Cristini, A.: The impact of Saharan dust on TMs solubility in rainwater in Sardinia, Italy, in: The Impact of Desert Dust Across the Mediterranean, edited by: Guerzoni, S. and Chester, R., Springer, Dordrecht, 11, 285–290, https://doi.org/10.1007/978-94-017-3354-0_28, 1996.*

---

## Author Comment (AC1)

This manuscript by Shelley et al. investigates aerosol trace element solubility and deposition fluxes over the polluted and dusty Mediterranean and Black Sea regions. The study provides valuable insights into the interplay between natural mineral dust and anthropogenic pollutants by analyzing aerosol samples to quantify the soluble and total concentrations of various lithogenic and anthropogenic elements. The authors explore how different air mass origins, such as North African dust and European pollution, influence these concentrations and examine the impact of atmospheric deposition on nutrient ratios and element budgets in the Mediterranean Sea.

While the observations are limited to a specific period, the paper nonetheless contributes significantly to understanding aerosol trace element solubility and its implications for marine nutrient cycling. Additionally, the manuscript is well-organized and clearly written. Below, I offer some minor suggestions for the authors consideration:

We thank the reviewer for their helpful comments. Our responses are given in blue text below, with changes to the manuscript in italics.

1. **Section 2.3.3**: Some ions mentioned in this section are not discussed in subsequent sections, such as $Br^-$ and $K^+$. I suggest removing these ions unless they are addressed later. Furthermore, is there data on $NH_4^+$ recovery that could be included?

   It is true that we do not discuss some of the ions mentioned here, but we prefer to leave them in the list so that others are aware that the data exist.
   Certified $NH_4^+$ concentrations are not available for ION-915 and KEJIM-02. Recoveries of $NH_4^+$ in our internal standards were within 5% of their target value (n = 4).

2. **Section 2.5**: As the authors acknowledge, there are substantial uncertainties in the estimation of deposition velocities (Vd). If this section is to remain in the main body of the text, I recommend providing a more detailed uncertainty analysis. For instance, could the authors incorporate model results to assess the impact of these uncertainties?

   We are not sure what type of modelling the reviewer had in mind, but we note that numerical models of atmospheric deposition are subject to very similar levels of uncertainty to those encountered in the calculations we have performed.
   In order to highlight the potential impact of the uncertainty in deposition velocities, we have added ranges to the values given for the potential contributions of atmospheric deposition to soluble element budgets in the Mediterranean. The text below also takes account of a related comment by Reviewer 2.

   *"Assuming that the aerosol samples collected during GA04 were representative of deposition to the region, mean dry deposition fluxes (Table 2) during summer (June - August) can account for 11 (3.6 − 32) % (Mn), 3.2 (1.1 − 9.6) % (Ni), 8.6 (2.9 − 26) % (Zn) and 1.0 (0.3 −*

*2.9) % (Cd) of the annual deficit in the surface budgets reported by Middag et al. (2022) in the western Mediterranean and 1.4 (0.5 – 4.3) % of the Ni deficit in the eastern basin (values in parentheses represent the range due to a 3-fold uncertainty in deposition velocity). The values for Ni in the eastern basin may be lower limits, as noted above."*

3. **Section 2.6**: The manuscript states that "samples were assigned to one of five air mass types, indicative of likely aerosol source characteristics as described below." Was this classification based on a subjective assessment, or was an algorithm or objective method used? Were there any ambiguous cases that the authors had to resolve, and if so, how?

   While tools such as openair (Carslaw and Ropkins, 2012) allow trajectories obtained from static locations to be classified using cluster analysis, we are not aware of any equivalent tools that can be applied when the trajectory origin point is moving (as is the case during cruises). From that point of view, our classification can be regarded as subjective to some extent. However, the broad classifications that we have used are quite distinct and there were very few cases of ambiguity in the classification. Most of those were related to dust transport, which can be affected by trajectories at multiple heights, but in those cases we had the physical characteristics (colour) of the samples as an additional aid to classification.

   We have added an additional figure to the Supplement (new Fig S1), which shows more detailed trajectories for the samples highlighted in Fig. 2, as well as adding text to the description of how air mass types were assigned (see response to Reviewer 2).

4. **Section 3.1**: In Figure 2, rather than showing an example trajectory for each air mass type, would it be possible to display the mean trajectory for each class? Additionally, the colors used for the borders of the LE and LAN areas are too similar to the trajectory colors, which makes the figure somewhat confusing. I suggest modifying the color scheme. Lastly, the text mentions 'LAM,' but the figure is labeled as 'LA'—this inconsistency should be corrected.

   We have modified the figure in the following ways. PSA regions are now shown using black patterns to differentiate them and reduce confusion with the colours used for trajectories. The missing "M" has been restored to "LAM". Thank you for spotting that. We have not opted to show mean trajectories because the ship's movement "disconnects" the means from the ship's track, which we do not think aids understanding. Further changes to the figure were made in response to comments from Reviewer 2. The revised figure can be seen in our response to those comments.

5. **Section 3.2**: In Figure 3 (and in similar figures later on), it would be clearer to differentiate air mass types by using distinct colors for each type rather than using different colors for the various elements, which does not seem to add much clarity to the interpretation.

   Thank you for this suggestion. We have modified the figures (3 – 6 and S1) as suggested. Figure 3 is shown as an example below. Note that in this example, low concentrations make

some of the coloured bars difficult to see, but this is not the case in the other figures. Since the order of the types is the same in all the figures, the desired information is still present.

[Figure]

Figure 3. Total concentrations of lithogenic elements ± 1 SD (pmol m⁻³), with enrichment factors relative to Al overlaid (circles). Unfilled bars indicate that analyte was below the limit of detection and bar represents 75% of the limit of detection. An EF was not determined if both Al and the element of interest were below the limit of detection. Up- / down-ward pointing grey arrows near EF markers indicate that values are minima / maxima because Al / the element were below the limit of detection. *Bars are coloured according to the air mass type of each sample*, blue = RNA, grey = WEU, orange = NAF, green = MED, pink = EEU. The dashed grey vertical lines indicate the legs of the cruise, with Leg 1-3 being left to right.

6. **Section 3.6**: The authors list several factors that influence element solubility. While they appropriately acknowledge that some mechanisms cannot be fully explored due to missing data, I believe that the discussion of acidity effects could be strengthened. Specifically, when discussing the role of acidic species concentrations, the authors should clarify to readers that while these concentrations provide useful insights, they do not directly represent aerosol pH. It would be helpful to explicitly state the limitations of using these parameters as proxies for aerosol acidity.

We agree that this makes a valuable addition to the discussion here. We have added the following text:

*"Ultimately, solubility enhancement through acid processing is dependent on the pH environment of the aerosol on an individual particle basis. This environment will vary strongly through the aerosol population, due to differences in internal mixing of acidic and alkaline species and trace elements (e.g. with particle size (Fang et al., 2017; Baker et al., 2020)). Furthermore, changes in the liquid water content of the particles (which is dependent on relative humidity and the hygroscopicity, and hence chemical composition, of the particles in question) can result in dramatic changes in pH, even when acid/alkaline ion balance varies little (Pye et al., 2020; Baker et al., 2021). Information about these factors is not available for the GA04 dataset and the insights provided by the above discussion of ion-solubility relationships are therefore limited."*

Additional References

Baker, A. R., Kanakidou, M., Nenes, A., Myriokefalitakis, S., Croot, P. L., Duce, R. A., Gao, Y., Ito, A., Jickells, T. D., Mahowald, N. M., Middag, R., Perron, M. M. G., Sarin, M. M., Shelley, R. U., and Turner, D. R.: Changing atmospheric acidity as a modulator of nutrient deposition and ocean biogeochemistry, Science Advances, 7, eabd8800, 10.1126/sciadv.abd8800, 2021.
Carslaw, D. C., and Ropkins, K.: openair — an R package for air quality data analysis, Environmental Modelling & Software, 27-28, 52–61, 2012.
Pye, H. O. T., Nenes, A., Alexander, B., Ault, A. P., Barth, M. C., Clegg, S. L., Collett Jr, J. L., Fahey, K. M., Hennigan, C. J., Herrmann, H., Kanakidou, M., Kelly, J. T., Ku, I. T., McNeill, V. F., Riemer, N., Schaefer, T., Shi, G., Tilgner, A., Walker, J. T., Wang, T., Weber, R., Xing, J., Zaveri, R. A., and Zuend, A.: The acidity of atmospheric particles and clouds, Atmospheric Chemistry and Physics, 20, 4809-4888, 10.5194/acp-2019-889, 2020.

---

## Author Comment (AC2)

**Review of "Aerosol trace element solubility and deposition fluxes over the polluted, dusty Mediterranean and Black Sea basins" by R. Shelley et al.**

Shelley et al. present a novel dataset of trace metals and P solubility and dry atmospheric fluxes from aerosols collected during GA04 cruise in Mediterranean (Med) and Black seas. It is interesting to have atmospheric data useful for studies on air-sea exchange in these regions which are influenced by various aerosol sources, notably with a gradient between East and West,.

The manuscript is well-written, logically organized, and adequately illustrated. Abstract is succinct and accurate. The discussion focuses in part on the comparison between the values found in the Med and Black Seas compared with the ones obtained in Atlantic Ocean, where the authors took part in several cruises. This comparison is interesting in order to situate the anthropogenic contribution in these two regions, which are not at the same distance from pollution sources. It would also have been useful to have a discussion in this context with the other measurements made in the Mediterranean, in particular for deposition fluxes. There is extremely little data available from campaigns at sea, but there are several observatories on islands with long-term measurements in these regions, e.g. Finokalia, Lampedusa or data from annual monitoring: Corsica, Balearic island…. A small putting in context and comparison with previous and recent works (since the date have been obtained in 2013) could be unvaluable to get the most out of these data.

Overall, the quality of the figures is not very good, but I think this is due to the changeover to pdf format and should be sorted out in the final manuscript.

We thank the reviewer for their helpful comments. Our responses are given in blue text below, with changes to the manuscript in italics.

**Specific comments:**

**Title:** Because of their location, the Mediterranean and the Black Sea are usually subject to polluted or desert dust-laden masses. In the case of this paper, no specific or intense events of desert dust or pollution have been reported. So I think the title could be simplified: "Aerosol trace element solubility and deposition fluxes over the Mediterranean and Black Sea basins"

We prefer to keep the title as it is. Our aim is not to suggest that intense pollution or dust events were sampled during the cruise, but to emphasise that the Mediterranean and Black Seas' atmospheres are subject to pollution and dust loads that are unusually high in the context of atmospheric conditions over the global ocean. This point may not be obvious to the broad readership of the journal.

**Figure 1**: In order to have a better view of the transect and the sampling zones, it would be useful to zoom in on the measurement area over sea (e.g., between 50°N and 25°N) and refine the thickness of the aerosol track.

It would be also useful to include the numbers of the filters that are noteworthy and discuss them in the text, for example, samples 18 and 20, which stand out in terms of soluble concentrations.

We have split the figure into 3 panels to show the cruise legs individually, and added some annotations to the map for leg 2, during which the east-bound and west-bound passages covered the same track. We have also added numbers to identify the samples indicated by the reviewer, as well as all of the samples for which example trajectories are shown in Fig. 2 (in order to address a later comment from the reviewer, without adding further complexity to Fig. 2).

[Figure]

*Figure 1. GA04 cruise track showing aerosol sampling periods as alternating thick black and grey bars for (a) Leg 1, (b) Leg 2 and (c) Leg 3. Note that the eastbound and westbound sections of Leg 2 followed very similar tracks and cannot be distinguished on this map. Numbers for samples referred to in the text and in Fig. 2 are indicated.*

**SC-01**. **Line 95-134**. The description of the field blanks (sampling and analysis) is clear, but there is no clear explanation regarding the use of the blank values. Are they subtracted from the total and soluble values? If so, which 'type' of blanks (1, 2, or 3)?

Good point. We have added the following text to address this comment:

"*Where blanks were above the analytical limit of detection for an analyte, these were averaged and subtracted from the results obtained for the samples. If blanks were below the limit of detection no blank subtraction was done.*"

**SC-02: Line 50:** "*For our flux calculations, we used vd values of 1 cm s-1 for the lithogenic elements and 0.1 cm s-1 for the anthropogenic elements*". The data show that certain elements are predominantly crustal during desert dust events (see EF values), whereas they are anthropogenic at other times. It would be interesting to calculate the percentage of desert dust inputs compared to anthropogenic inputs during the campaign. If it is significant, in order to better constrain the flux calculations, shouldn't these dust events (i.e. vd values of lithogenic) be taken into account in the flux calculations for these elements?

This is an interesting suggestion, but we have insufficient information to calculate the relative contributions of lithogenic and anthropogenic sources for the elements affected in this way. There is also the further complication that we do not know the size distribution of the elements, or of their soluble fractions (size distribution being the major determinant of deposition velocity). We therefore do not think that we can address this comment in a quantitative way.

Low EF values for elements designated as anthropogenic (and therefore assumed to have low deposition velocities) occur chiefly during Leg 1 in the eastern Mediterranean (for P, V, Ni, Cu and Zn) and Leg 2 (for V and Ni). We have amended the text to note that the respective values in Table 2 (and the contributions to the Middag et al. budget deficits) may be underestimates. Since we report deposition fluxes as regional averages (rather than for individual samples) any underestimation will not be as large as implied by the low EF values for individual samples.

We have added the following text:

"*Note that the low EF values observed for some "anthropogenic" elements (e.g. P, V, Ni, Cu and Zn for some samples during Leg 1 in the eastern Mediterranean, and V and Ni during Leg 2; Fig. 5) may imply a higher proportion of the soluble fractions of these elements in coarse aerosols than in other samples encountered during GA04. This, in turn, may suggest that deposition velocities in these cases might be higher than the value (0.1 cm s$^{-1}$) used in our calculations. Thus, deposition fluxes in these cases may be higher than those given in Table 2, although the absence of aerosol size distribution data for these soluble elements makes the magnitude of such underestimation difficult to quantify.*"

**SC-03: 3.1. Air mass origins during GA04 :**

The analysis of back-trajectories is useful for classifying the different samples based on potential sources, but a more detailed analysis of the actual sources might help better understand the various influences in the discussion section. To aid data analysis in the case of desert dust arrivals, it would be interesting to examine dust emissions, using observations such as RGB SEVIRI, along the back-trajectories, rather than relying on PSAs which do not provide information on the actual origin of the transported dust. Indeed, just because an air mass passes over potentially emitting areas does not mean it picks up desert dust; the winds must be strong enough to trigger dust production. For example, in the case of sample 38, the air mass passes over both Tunisia and Algeria, which are two emissive sources with quite different mineralogical characteristics. The Tunisian source is richer in calcite, which can play a role in neutralizing acids (Kandler et al., 2007). It coud be invaluable to consider this point in the discussion of acid effect on solubility.

*Kandler, K., Benker, N., Bundke, U., Cuevas, E., Ebert, M. and co-authors. 2007. Chemical composition and complex refractive index of Saharan Mineral Dust at Izaña, Tenerife (Spain) derived by electron microscopy. Atmos. Environ. 41, 8058–8074.*

Dust source is not the focus of our paper. We added PSAs to Fig. 2 for illustrative purposes only. In the context of our work, the most significant information is that some of the samples collected contained Saharan dust. We have added text to Section 3.1 to state that we have not attempted to identify individual dust sources and to emphasise that the presence of dust was confirmed by the colour of the samples themselves (as already noted on line 180 of the original manuscript), as well as by back trajectory analysis.

"*As illustrated in Fig. 2, trajectories for this group passed over regions identified as potential source areas (PSAs) for dust aerosols, notably the 'Libya/Egypt' and 'Libya/Algeria/Mali' PSAs (Guinoiseau et al., 2022), but we do not attempt to assign potential dust sources for individual samples. All of the NAF samples showed the orange colouration associated with desert dust.*"

We address the issue of Ca content in the GA04 samples in relation to a later comment from the reviewer.

**Figure 2:** This figure is complex enough. To clarify Figure 2, perhaps using only the back-trajectories at 10 meters might be sufficient since the collection takes place at the height of the ship? Furthermore, the text does not explain which altitude (10, 500, or 1000 meters) is used as the reference for classifying the air masses. It would be useful to specify, if possible, the altitudes of the boundary layer during the campaign, as the back-trajectories at 1000 meters are only relevant if the boundary layer reaches at least 1 km and is well mixed during the sampling?

For most of the air mass types encountered, surface level transport is indeed the most important factor. However, this is not the case for Saharan dust transport (Scerri et al., 2016) for which high altitude transport, including above the boundary layer, is also important. We have added text to Section 2.6 to better explain how we classified samples according to their air mass back trajectories:

*"Assignments were done principally based on the surface level (10 m) trajectories. However, higher altitude transport (up to 3000 m) can be significant for Saharan dust over the Mediterranean (Scerri et al., 2016), so upper levels were also considered in the context of transport from North Africa."*

In order to simplify the figure, we have removed the upper-level trajectories for all but the NAF-type samples.

[Figure]

*Figure 2. GA04 cruise track showing the classification of aerosol samples into 5 air mass types. Example AMBTs (and sample numbers) for each type are shown for arrivals at 10 m (dashed lines) and (for NAF only) 1000 m (dotted lines). The Libya-Algeria-Mali (LAM) and Libya-Egypt (LE) Potential Source Areas (Guinoiseau et al., 2022) are also shown.*

Could you add the filter numbers at the beginning and end of the back-trajectory types to Figure 2 to link this figure with the subsequent ones, especially in the area where the routes of legs 1 and 3 overlap?

Rather than add extra information to this figure (which the reviewer notes is already complex), we have added filter numbers to Fig. 1 to address this comment.

In the same idea, please color the bars in Figures 3, 4, 5, and 6 according to the air mass origins (rather than just the colored horizontal bar at the top of each panel). This would make it much easier for the reader to link the sample to its origin (which is the basis of your data analysis in the following sections). Since the names of the metals are labeled on each panel, it should not cause any confusion.

Thank you for the suggestion. This has been done (see response to Reviewer 1).

**SC-04 : Lithogenic elements**

**L180** : Please indicate in the text the average value of metal concentrations (e.g., iron) in the NAF and non-NAF samples, so that these values can be used as a reference for metal concentrations according to their origin, and to quantify the contribution of Saharan air masses to the background metal levels during the campaign

We have added a table to the supplementary information (new Table S2, with subsequent supplementary tables renumbered), summarising the characteristics of each air mass type (median and range concentrations) for both soluble and total metals.

*"For all the primarily lithogenic elements, the influence of Saharan dust (NAF air mass type) on the aerosol total metal concentrations is clearly visible in samples TM8-14 (Fig. 3, Table S2)."*

**L200 and 201** : I don't understand the connection between the first sentence and the given example since s-Fe concentrations in the RNA samples are lower than in the NAF samples.

We thank the reviewer for spotting this. The text has been reordered to make the link to the first sentence more obvious:

*"For example, median s-Fe concentrations in aerosols of the Mediterranean and European air mass types (110, 320 and 180 pmol m$^{-3}$ for MED, WEU, EEU types respectively) were all considerably higher than in NAF aerosols (54 pmol m$^{-3}$) during GA04. The median s-Fe concentration in the RNA air mass was ~30 pmol m$^{-3}$, similar to the value previously reported for this air mass type in the remote Atlantic: ~15 pmol m$^{-3}$ (Baker and Jickells, 2017)."*

**L200 to 205** : In addition to the s-Fe values, include the average solubility value. This will help illustrate your discussion on the relationship between solubility and s-Fe concentrations.

We have added example solubility values as follows:

*"(e.g. median fractional solubilities for Fe in the NAF, MED, WEU, EEU and RNA types during GA04 were 0.4, 2.9, 12.4, 2.8 and 5.1 %, respectively)"*

**SC-05 : Anthropogenic elements**

**L219** : add ", except P" before Fig.5

We do not think that this change is necessary. The sentence referred to is clear that the statement does not apply to all elements and the exception (P) is obvious from the figure.

**L223 then 242**: *"Soluble V and Ni concentrations were strongly correlated (r2 = 0.973, p = <<0.01),*

*which is consistent with V and Ni in aerosols being markers of shipping emissions, and these emissions being the dominant control on s-V and s-Ni concentrations (Becagli et al., 2012; Baker and Jickells, 2017)."*

Here, I think that it will be more precise to write: "Soluble V and Ni concentrations were strongly correlated ($r^2$ = 0.973, p << 0.01). V and Ni in aerosols being primarily attributed to shipping

emissions (Becagli et al., 2012), this is consistent with the idea that these emissions are the dominant factor controlling s-V and s-Ni concentrations, as previously observed in the Atlantic Ocean (Baker and Jickells, 2017)."

In order to reinforce this conclusion, is there a correlation between the t-Ni and t-V values in the non-NAF samples that could confirm a source related to maritime traffic, as observed by Becagli et al. (2012)? It could be interesting to confirm (or not) the Becagli et al.'s conclusion obtained on local point in Med, on a large area of Med and Black seas. If it is not the case, the fact that s-Ni and s-V are correlated indicate that even if these metals can have different anthropogenic sources (that is highly possible in Med and consistent with your sentence L223-225), the 'shipping' source, is the primary contributor of soluble atmospheric inputs for these metals.

Total Ni and V concentrations are indeed very strongly correlated in both the non-NAF and NAF samples collected. We have changed this text as suggested above, with a slight modification to take account of the reviewer's second point.

"Soluble (and total) V and Ni concentrations were strongly correlated ($r^2$ = 0.973 (soluble) and $r^2$ = 0.955 (total), both p = <<0.01). V and Ni in aerosols at Lampedusa in the central Mediterranean being primarily attributed to shipping emissions (Becagli et al., 2012), this is consistent with the idea that these emissions are the dominant factor controlling s-V and s-Ni concentrations, as previously observed in the Atlantic Ocean (Baker and Jickells, 2017)."

**L 225**: "*suggesting that P (as well as V, Ni, Cu, Zn and Pb) has important non-dust sources*": P could be issued from European dust source related to agricultural soils enriched in P by fertilizers. That could explain that t-P follows the behavior of lithogenic elements, even if the EF are >10 (see e.g. Bergametti data in Mahowald et al. (2008)).

*Mahowald, N., et al. (2008), Global distribution of atmospheric phosphorus sources, concentrations and deposition rates, and anthropogenic impacts, Global Biogeochem. Cycles, 22, GB4026, doi:10.1029/2008GB003240*

We have altered the sentence to acknowledge the possibility of P enrichment via agricultural fertilizer application.

"..., suggesting that P (as well as V, Ni, Cu, Zn and Pb) has important non-natural dust sources (note that Mahowald et al. (2008) suggest that P enrichment might occur through the uplift of artificially fertilized soils)."

**L233-235** : It would be more correct to write : «... *that the Mediterranean and Black Sea atmospheres were still subject to significant anthropogenic influence of Pb emissions sources in 2013"*, since these data are more than 10 years old (European anthropogenic Pb emissions decreased since the cruise, see e.g. annual European Union emission inventory report 1990-2022) and the emissions come from the proximity of various anthropogenic sources compared with the Atlantic.

We agree that it is a good idea to remind the reader when the samples were collected and have changed the end of this sentence to "... *suggesting that the Mediterranean and Black Sea atmosphere were still subject to significant anthropogenic emissions of Pb in 2013.*"

**Figure 4** : Authors explained that some values of solubility reach 300% due to probable contaminations. I am rather sceptical about the presentation of these values in the same way as the other values, since it's impossible to estimate the contamination and these values suggest that the solubility is high when it may only be a few %, for example. I think it would be clearer to show these values in "transparent" colour, rather than in the same colour as the uncontaminated values.

We have changed the colour of the affected symbols to grey in line with this suggestion.

**SC-06 : Influence on element solubility**

**L298-309**: Your discussion is based on the acid effect from polluted species, but your dissolution protocol is also carried out in a buffered medium. To what extent could the excess acid affect the values you find?

The high concentration of the buffer (1 M) means that the acids contained in the samples (which contribute nitrate and sulfate concentrations in the leach solutions of much less than 1 mM) do not impact the results. The buffered leach medium therefore provides solubility values that are consistent across the dataset.

Another point that raises questions in this part of the discussion, although the concentrations of inorganic acid and base species are higher than in the Atlantic, it is the capacity of these species to acidify or neutralise the aerosol, and therefore play on the pH, that will affect solubility. For this, it is not so much the concentrations that are important, but the balance of neutralisation of the inorganic (and organic) acidic species by the alkaline species (or vice versa). You use the $nssSO_4$/Fe ratio to estimate the excess of acid over Fe, but this is only valid if $nssSO_4$ is not neutralised by $NH_4^+$ or by the presence of calcite, for example, in your samples. In your case, as you do not have Ca values, you cannot estimate the neutralisation capacity linked to the presence of calcite in NAF samples, but you can estimate the effect of $NH_4^+$ in all your samples. It would be more accurate to use, for example, the neutralisation ratio (Silvern et al., 2017) rather than the $nssSO_4$/Fe value.

Thank you for this comment. The potential for calcite to titrate acidic species during atmospheric processing is also potentially important. We do have soluble Ca data for the GA04 samples (although not for Leg 2) and also for the Atlantic dataset. Non-seasalt Ca concentrations (as a proportion of total Fe concentrations) during GA04 were in the upper range of values encountered in the Atlantic data (nss-Ca:t-Fe = 1.4-10.1 mol mol$^{-1}$ for NAF / GA04 and 0.07-10.8 mol mol$^{-1}$ for SAH / Atlantic), but not outside the range of the Atlantic data. Thus, it does not appear that differences in Ca content between the two datasets can account for the differences in solubility observed between the two datasets.

We have added the following comment to the text:

"*Calcite content varies significantly with dust source (generally being higher in sources in the north of the Sahara (Chiapello et al., 1997; Kandler et al., 2007)) and may also impact solubility enhancement through neutralisation of acidity. However, while the nss-$Ca^{2+}$ / t-Fe ratio in the NAF samples (1.4-10.1 mol mol$^{-1}$) was relatively high, it was not outside the range of values for this ratio for Saharan dust in the Atlantic dataset (0.07-10.8 mol mol$^{-1}$). Variations in calcite content therefore seem unlikely to account for the observed differences between solubility over the Atlantic and Mediterranean.*"

We do not think that ion balance calculations are helpful in the interpretation of this dataset. See response to Reviewer 1.

*Silvern, R. F., Jacob, D. J., Kim, P. S., Marais, E. A., Turner, J. R., Campuzano-Jost, P., and Jimenez, J. L.: Inconsistency of ammonium–sulfate aerosol ratios with thermodynamic models in the eastern US: a possible role of organic aerosol, Atmos. Chem. Phys., 17, 5107–5118, https://doi.org/10.5194/acp-17-5107-2017, 2017.*

**L314** : *"Overall, the low solubility of Al, Mn, Fe, Co and Th in NAF samples (relative to similar samples collected over the Atlantic; Fig. 7) appears to be related to the short atmospheric transport pathway to the Mediterranean."* : The low solubility 'could be' related, this is one possible explanations but it would also be interesting to discuss the effect of other phenomena to explain this low solubility compared to the values in the Atlantic. First of all, the dust emitting sources are different between the samples taken in the Mediterranean (North of Africa) and those taken in the Atlantic (Morocco, Central Sahara, Bodélé, Sahel, etc.). The difference in sources (including the difference in mineralogy, particle size distribution and calcite load, and therefore pH) could also explain the differences you observe. You seem to consider that all Saharan emission sources have the same solubility (and should respond in the same way to the process along the transport route), but the data at the source show that the solubility of iron in particular can vary from one source to another (see e.g. Shi et al., 2011; Paris et al., 2010). In particular, the solubility in the first dust period appears to be lower than in the second NAF period, even though the route is comparable.

We do not think that dust sources to the Mediterranean and Atlantic are as distinct as the reviewer's comment implies. The more northerly (high calcite) sources are also transported to the west, as observed in several studies (e.g. Kandler et al., 2007, Chiapello et al., 1997).

We do not have sufficient evidence to establish the geographic distribution of the dust sources for individual samples, nor the mineralogy or trace element solubilities of those sources (if we were able to identify them). We do not make any assumptions about the solubility of the various sources of dust from their point of origin, whether that dust was sampled over the Mediterranean or the Atlantic. Our discussion is only based on the observed properties of the dust when collected at sea. The chemical characteristics of individual Saharan dust sources are homogenised during long-range transport (Trapp et al., 2010) and similar mixing of dust sources is also possible during transport to the Mediterranean.

Moreover, unless I'm mistaken, most of the measurements made in the Atlantic were taken during a different period than the GA04 cruise and the seasonality of the sources is different. It would be interesting to specify this in the text, particularly as photochemical processes are probably accentuated in summer in the Med and the Black Sea. This could perhaps explain the differences in solubility in the case of non-NAF samples, e.g. linked to oxidation processes or to the greater presence of organic matter or potential mixing with anthropogenic forms.

The reviewer is correct that most of the Atlantic data shown in Fig. 7 were not collected during the summer months. However, our Atlantic database is large and there are in fact slightly more samples from June – August in the Atlantic dataset than were collected during GA04. Seasonality is therefore unlikely to account for the observed behaviour.

As noted in previous studies (e.g. Baker and Croot, 2010), the broadscale similarities between the variations in the solubility of Fe and Al (Fig. 7g & a) rule out oxidation processes as the dominant factor explaining the differences in solubility within the GA04 dataset or between the

GA04 data and previous observations over the Atlantic. Aluminium has no redox chemistry, yet its solubility behaviour is very similar to that of Fe.

We have added the following comment (new text in italics):

"Short atmospheric transport times also reduce the potential for solubility of some elements to be enhanced through photochemical redox changes (e.g. insoluble Fe (III) to soluble Fe (II); Longo et al., 2016), *although the very similar solubility behaviour of Fe and Al imply that redox changes are not a major control on Fe solubility (since Al has no redox chemistry)*."

*Shi, Z., et al. (2011), Influence of chemical weathering and aging of iron oxides on the potential iron solubility of Saharan dust during simulated atmospheric processing, Global Biogeochem. Cycles, 25, GB2010, doi:10.1029/2010GB003837*

*Paris, R., Desboeufs, K. V., Formenti, P., Nava, S., and Chou, C.: Chemical characterisation of iron in dust and biomass burning aerosols during AMMA-SOP0/DABEX: implication for iron solubility, Atmos. Chem. Phys., 10, 4273–4282, https://doi.org/10.5194/acp-10-4273-2010, 2010.*

**SC-07 Dry deposition**

**L 331: "Dry deposition therefore probably accounts for the majority of total atmospheric fluxes to the basins over the study period."** To reach this conclusion, rather than compare with the rainfall, it could be interesting to compare with the elemental fluxes obtained by wet or total deposition measurements in the same period in these areas. For example, if we consider an iron concentration of 340 nmol.L-1 in rain collected in June in the western Mediterranean (Desboeufs et al., 2022, it is just an example as base for the calculation ) with a rainfall of 1 mm.d-1, this is equivalent to a flux of 340 nmol.m-2.d-1, i.e. in the average of values obtained here.

This is an interesting suggestion, but we do not think that it is possible to make a meaningful comparison to wet deposition with the data available. Desboeufs et al. (2022) report an iron concentration for one sample collected during a single rainfall event in the western Mediterranean (and one other sample collected in the eastern basin). The western event (whose Fe concentration is mentioned above) was reported to contain significant amounts of Saharan dust and is unlikely to be representative of wet deposition to the western Mediterranean basin as a whole. We have added a comment to emphasise the difficulties in attempting to estimate wet fluxes.

"..., *although rainfall composition data over the open Mediterranean are extremely scarce (Desboeufs et al. 2022) so this is difficult to verify*."

*Desboeufs, K., et al.: Wet deposition in the remote western and central Mediterranean as a source of trace metals to surface seawater, Atmos. Chem. Phys., 22, 2309–2332, https://doi.org/10.5194/acp-22-2309-2022, 2022.*

Only Guieu's work is cited, even though it dates from the early 2000s and several studies on deposition have since been carried out in these regions. Numerous measurements have been made in the Mediterranean and there is a great deal of data available (e.g. the long-term measurements in Finokalia, the literature on the total deposition in Sardinia, Corsica or Balearics Islands ). It is a pity that no comparison is provided with these data obtained on

islands to see the potential differences with our data, notably about the source of trace metals and nutrients, which are studied in this literature, e.g.:

We focused comparisons to previous studies on those which reported aerosol concentrations because these works gave values which were directly comparable to our measurements. Not only are there substantial uncertainties in calculating dry deposition flux from aerosol concentration (as we discussed), but there are also significant limitations in the sampling devices employed to measure deposition fluxes at island and coastal sites (e.g. in many of the studies cited by the reviewer below). Most significant of these is that these sampling devices cannot reproduce the varying aerosol deposition conditions over the ocean that occur with changes in sea state at different wind speeds (Slinn and Slinn, 1980).

Nevertheless, we have added comparisons to previously published estimates of deposition fluxes in response to this comment and a comment from Reviewer 3, although we were not able to find directly comparable or relevant values in all of the sources suggested by this reviewer (some of these report total trace metal fluxes, but not the soluble fluxes given in our work).

"*However, our estimates appear broadly comparable with fluxes reported from longer-term sampling around the Mediterranean when annual averages are expressed on a daily basis (e.g. DIN 70-126 $\mu$mol m$^{-2}$ d$^{-1}$ (western basin) and 79-210 $\mu$mol m$^{-2}$ d$^{-1}$ (eastern basin), (both Markaki et al., 2010) and 166 $\mu$mol m$^{-2}$ d$^{-1}$ (Crete) (Theodosi et al., 2019); dust 35 mg m$^{-2}$ d$^{-1}$ (Crete) (Theodosi et al., 2019) and 3.8-5.3 mg m$^{-2}$ d$^{-1}$ (Corsica) (Desboeufs et al., 2018). Our estimate of t-Al (dust) deposition to the Black Sea is also similar to the value reported by Theodosi et al. (2013) based on sampling at two coastal sites (~ 6 mg dust m$^{-2}$ d$^{-1}$, assuming Al is 8 % of dust by mass).*"

*Theodosi C., Markaki Z., Pantazoglou F., Tselepides A., Mihalopoulos N., Chemical composition of downward fluxes in the Cretan Sea (Eastern Mediterranean) and possible link to atmospheric deposition: A 7 year survey, Deep-Sea Research Part II, 164, 89-99, 2019.*

*Desboeufs, K., Bon Nguyen, E., Chevaillier, S., Triquet, S., and Dulac, F.: Fluxes and sources of nutrient and trace metal atmospheric deposition in the northwestern Mediterranean, Atmos. Chem. Phys., 18, 14477–14492, doi.org/10.5194/acp-18-14477-2018, 2018.*

*Christodoulaki S., G. Petihakis, N. Mihalopoulos, K. Tsiaras, G. Triantafyllou, M. Kanakidou, Human-Driven Atmospheric Deposition of N and P Ccontrols on the East Mediterranean Marine Ecosystem, JAS, 73, 1611- 1619, 2016.*

*Kanakidou M., S. Myriokefalitakis, N. Daskalakis, G. Fanourgakis, A. Nenes, A. Baker, K. Tsigaridis, N. Mihalopoulos, Past, Present and Future Atmospheric Nitrogen Deposition, JAS, 73, 2039-2047, 2016.*

*Longo, A. F., Ingall, E. D., Diaz, J. M., Oakes, M., King, L. E., Nenes, A., Mihalopoulos, N., Violaki, K., Avila, A., Benitez-Nelson, C. R., Brandes, J., McNulty, I., and Vine, D. J.: P-NEXFS analysis of aerosol phosphorus delivered to the Mediterranean Sea, Geophys. Res. Lett., 41, 4043–4049, https://doi.org/10.1002/2014GL060555, 2014.*

*Im U., S. Christodoulaki, K. Violaki, P. Zarbas, M. Kocak, N. Daskalakis, N. Mihalopoulos and M. Kanakidou, Atmospheric deposition of nitrogen and sulfur over Europe with focus on the Mediterranean and the Black Sea, Atmospheric Environment, 81, 660-670, 2013.*

*Markaki Z., M.D. Loye-Pilot, K. Violaki, L. Benyahya, N. Mihalopoulos, Variability of atmospheric deposition of dissolved nitrogen and phosphorus in the Mediterranean and possible link to the anomalous seawater N/P ratio, Marine Chemistry, Volume 120, Issues 1-4, Pages 187-194, 2010.*

*Theodosi C., Z. Markaki, A. Tselepides, N. Mihalopoulos, The significance of atmospheric inputs of soluble and particulate major and trace metals to the eastern Mediterranean seawater, Marine Chemistry, Volume 120, Issues 1-4, 20, 154-163, 2010.*

*Theodosi C., Z. Markaki, N. Mihalopoulos, Iron speciation, solubility and temporal variability in wet and dry deposition in the Eastern Mediterranean, Marine Chemistry, Volume 120, Issues 1-4, 20, 100-107, 2010.*

*Guerzoni, S., Molinaroli, E., Rossini, P., Rampazzo, G., Quarantotto, G., and Cristini, S.: Role of desert aerosol in metal fluxes in the Mediterranean area, Chemosphere, 39, 229–246, https://doi.org/10.1016/S0045-6535(99)00105-8, 1999.*

*Frau, F., Caboi, R., and Cristini, A.: The impact of Saharan dust on TMs solubility in rainwater in Sardinia, Italy, in: The Impact of Desert Dust Across the Mediterranean, edited by: Guerzoni, S. and Chester, R., Springer, Dordrecht, 11, 285–290, https://doi.org/10.1007/978-94-017-3354-0_28, 1996.*

**Additional References**

Baker, A. R., and Croot, P. L.: Atmospheric and marine controls on aerosol iron solubility in seawater, Marine Chemistry, 120, 4-13, 10.1016/j.marchem.2008.09.003, 2010.

Chiapello, I., Bergametti, G., Chatenet, B., Bousquet, P., Dulac, F., and Santos Soares, E.: Origins of African dust transported over the northeastern tropical Atlantic, Journal of Geophysical Research, 102, 13701-13709, 10.1029/97JD00259, 1997.

Scerri, M. M., Kandler, K., and Weinbruch, S.: Disentangling the contribution of Saharan dust and marine aerosol to $PM_{10}$ levels in the Central Mediterranean, Atmospheric Environment, 147, 395-408, 10.1016/j.atmosenv.2016.10.028, 2016.

Slinn, S. A., and Slinn, W. G. N.: Predictions for particle deposition on natural waters, Atmospheric Environment, 14, 1013-1016, 1980.

Theodosi, C., Stavrakakis, S., Koulaki, F., Stavrakaki, I., Moncheva, S., Papathanasiou, E., Sanchez-Vidal, A., Koçak, M., and Mihalopoulos, N.: The significance of atmospheric inputs of major and trace metals to the Black Sea, Journal of Marine Systems, 109-110, 94-102, 10.1016/j.jmarsys.2012.02.016, 2013.

Trapp, J. M., Millero, F. J., and Prospero, J. M.: Temporal variability of the elemental composition of African dust measured in Trade Wind aerosols at Barbados and Miami, Marine Chemistry, 120, 71-82, 10.1016/j.marchem.2008.10.004, 2010.

---

## Author Comment (AC3)

**Review of "Aerosol trace element solubility and deposition fluxes over the polluted, dusty Mediterranean and Black Sea basins" by R. Shelley et al.**

An interesting work reporting trace metals, their solubility and dry atmospheric fluxes during GA04 cruise in Mediterranean (Med) and Black seas. The manuscript is well-written and deserves publication after addressing the following points:

We thank the reviewer for their helpful comments. Our responses are given in blue text below, with changes to the manuscript in italics.

General comments:

- Since the campaign covers a period during summer, as title I would suggest: Aerosol trace element solubility and deposition fluxes over the Mediterranean and Black Sea basins during summer time (or warm period if they wish).

  We do not think that this change is necessary. The season during which the cruise took place is already mentioned in the first sentence of the Abstract.

- There is a good amount of works based on either long-term or annual basis sampling and reported fluxes of nutrients and elements in the Mediterranean and Black Sea which are missing. Please compare with them. Below a non-exhaustive list:

Theodosi C., Markaki Z., Pantazoglou F., Tselepides A., Mihalopoulos N., Chemical composition of downward fluxes in the Cretan Sea (Eastern Mediterranean) and possible link to atmospheric deposition: A 7 year survey, Deep-Sea Research Part II, 164, 89-99, 2019.

Desboeufs, K., Bon Nguyen, E., Chevaillier, S., Triquet, S., and Dulac, F.: Fluxes and sources of nutrient and trace metal atmospheric deposition in the northwestern Mediterranean, Atmos. Chem. Phys., 18, 14477–14492, doi.org/10.5194/acp-18-14477-2018, 2018.

Im U., S. Christodoulaki, K. Violaki, P. Zarbas, M. Kocak, N. Daskalakis, N. Mihalopoulos and M. Kanakidou, Atmospheric deposition of nitrogen and sulfur over Europe with focus on the Mediterranean and the Black Sea, Atmospheric Environment, 81, 660-670, 2013.

Markaki Z., M.D. Loye-Pilot, K. Violaki, L. Benyahya, N. Mihalopoulos, Variability of atmospheric deposition of dissolved nitrogen and phosphorus in the Mediterranean and possible link to the anomalous seawater N/P ratio, Marine Chemistry, Volume 120, Issues 1-4, Pages 187-194, 2010.

Theodosi C., Z. Markaki, A. Tselepides, N. Mihalopoulos, The significance of atmospheric inputs of soluble and particulate major and trace metals to the eastern Mediterranean seawater, Marine Chemistry, Volume 120, Issues 1-4, 20, 154-163, 2010.

Guerzoni, S., Molinaroli, E., Rossini, P., Rampazzo, G., Quarantotto, G., and Cristini, S.: Role of desert aerosol in metal fluxes in the Mediterranean area, Chemosphere, 39, 229–246, https://doi.org/10.1016/S0045-6535(99)00105-8, 1999.

> Thank you. These papers were also mentioned by reviewer 2. We have included relevant comparisons where possible. See our response to Reviewer 2.

For Black Sea, Kocak et al., 2014 and references therein:  Atmospheric deposition of macronutrients (dissolved inorganic nitrogen and phosphorous) onto the Black Sea and implications on marine productivity, Journal of the Atmospheric Sciences 73 (4), 1727-1739, 2014. There total deposition measurements performed in Black Sea (Varna, Bulgaria) during similar period (2013-2014) were reported

> Since we were not able to collect samples for dissolved inorganic nitrogen and phosphorus analysis during the Black Sea leg of GA04, this paper does not provide any information relevant to our manuscript. Examination of the papers cited by Kocak et al. did not provide many flux determinations comparable to those we report, but we did find total Al flux values in Theodosi et al. (2013), which we were able to compare to the values in Table 2. Our detailed response to this comment is included in our response to Reviewer 2.

Other comments:

- Few words on meteorological conditions are missing. For instance, any rain event occurred during the cruise? Wind speed variability? This information is valuable for the reader.

  > Unfortunately, we are not able to provide this information.

- Figure 4: How the authors explain the increased soluble levels and the extremely high solubility for terrigenic elements such as Al, Fe and especially Mn (almost up to 100% for the last) under the influence of EEU air masses (pink color). A short comment would be very useful.

  > We do not have anything to add to the comments that we already made on this (lines 320-324 of the original manuscript).

- Figure 7: Solubility as a function of aerosol load. Similar figure and results were reported by Theodosi et al., 2010 at Finokalia Crete for both wet and dry deposition. Please compare your findings with this work.

  We have added a specific reference to this dataset. However, it is not directly comparable to the GA04 data, since it is derived from flux measurements and used a different method to determine the soluble fraction of Fe.

- For solubility: Did the authors calculate ionic balance as a better index of acidity or even better (SO4+NO3/NH4) ratio (in equivalent)? Any relation of (SO4+NO3/NH4) ratio with solubility?

  We do not think that ion balance calculations are helpful in the interpretation of this dataset. See response to Reviewer 1.

- N/P ratio in the Mediterranean: Your finding about N/P variability across Mediterranean is in very good agreement with the study of Markaki et al 2010 and I think it is worth mentioning. "Markaki Z., M.D. Loye-Pilot, K. Violaki, L. Benyahya, N. Mihalopoulos, Variability of atmospheric deposition of dissolved nitrogen and phosphorus in the Mediterranean and possible link to the anomalous seawater N/P ratio, Marine Chemistry, Volume 120, Issues 1-4, Pages 187-194, 2010"

  We have added the citation, as suggested.

- For Black sea fluxes compare with the work of Kocak et al., 2014 and references therein.

  See above.